# DIB-R++: Learning to Predict Lighting and Material with a Hybrid Differentiable Renderer

**Wenzheng Chen**[1,2,3]     **Joey Litalien**[4,*]     **Jun Gao**[1,2,3]     **Zian Wang**[1,2,3]

**Clement Fuji Tsang**[1]     **Sameh Khamis**[1]     **Or Litany**[1]     **Sanja Fidler**[1,2,3]

NVIDIA[1]     University of Toronto[2]     Vector Institute[3]     McGill University[4]

{wenzchen, jung, zianw, cfujitsang, skhamis, olitany, sfidler}@nvidia.com, joey.litalien@mail.mcgill.ca

## Abstract

We consider the challenging problem of predicting intrinsic object properties from a single image by exploiting differentiable renderers. Many previous learning-based approaches for inverse graphics adopt rasterization-based renderers and assume naive lighting and material models, which often fail to account for non-Lambertian, specular reflections commonly observed in the wild. In this work, we propose DIB-R++, a hybrid differentiable renderer which supports these photorealistic effects by combining rasterization and ray-tracing, taking the advantage of their respective strengths—speed and realism. Our renderer incorporates environmental lighting and spatially-varying material models to efficiently approximate light transport, either through direct estimation or via spherical basis functions. Compared to more advanced physics-based differentiable renderers leveraging path tracing, DIB-R++ is highly performant due to its compact and expressive shading model, which enables easy integration with learning frameworks for geometry, reflectance and lighting prediction from a single image without requiring any ground-truth. We experimentally demonstrate that our approach achieves superior material and lighting disentanglement on synthetic and real data compared to existing rasterization-based approaches and showcase several artistic applications including material editing and relighting.

## 1   Introduction

Inferring intrinsic 3D properties from 2D images is a long-standing goal of computer vision [3]. In recent years, differentiable rendering has shown great promise in estimating shape, reflectance and illumination from real photographs. Differentiable renderers have become natural candidates for learning-based inverse rendering applications, where image synthesis algorithms and neural networks can be jointly optimized to model physical aspects of objects from posed images, either by leveraging strong data priors or by directly modeling the interactions between light and surfaces.

Not all differentiable renderers are made equal. On the one hand, recent physics-based differentiable rendering techniques [31, 45, 2, 44, 59] try to model the full light transport with proper visibility gradients, but they tend to require careful initialization of scene parameters and typically exhibit high computational cost which limits their usage in larger end-to-end learning pipelines. On the other hand, performance-oriented differentiable renderers [10, 27, 36, 25] trade physical accuracy for scalability and speed by approximating scene elements through neural representations or by employing simpler shading models. While the latter line of work has proven to be successful in 3D scene reconstruction,

---

*Work done during an internship at NVIDIA.

[1]Project page: https://nv-tlabs.github.io/DIBRPlus.

35th Conference on Neural Information Processing Systems (NeurIPS 2021).

the frequent assumptions of Lambertian-only surfaces and low-frequency lighting prevent these works from modeling more complex specular transport commonly observed in the real world.

In this work, we consider the problem of *single-view 3D object reconstruction without any 3D supervision*. To this end, we propose DIB-R++, a hybrid differentiable renderer that combines rasterization and ray-tracing through an efficient deferred rendering framework. Our framework builds on top of DIB-R [10] and integrates physics-based lighting and material models to capture challenging non-Lambertian reflectance under unknown poses and illumination. Our method is versatile and supports both single-bounce ray-tracing and a spherical Gaussian representation for a compact approximation of direct illumination, allowing us to adapt and tune the shading model based on the radiometric complexity of the scene.

We validate our technique on both synthetic and real images and demonstrate superior performance on reconstructing realistic materials BRDFs and lighting configurations over prior rasterization-based methods. We then follow the setting proposed in Zhang et al. [62] to show that DIB-R++ can reconstruct scene intrinsics also from real images without any 3D supervision. We further apply our framework to single-image appearance manipulation such as material editing and scene relighting.

## 2   Related Work

**Differentiable Rendering.** Research on differentiable rendering can be divided into two categories: physics-based methods focusing on photorealistic image quality, and approximation methods aiming at higher performance. The former differentiates the forward light transport simulation [31, 45, 2, 44, 59] with careful handling of geometric discontinuities. While capable of supporting global illumination, these techniques tend to be relatively slow to optimize or require a detailed initial description of the input in terms of geometry, materials, lighting and camera, which prevents their deployment in the wild. The latter line of works leverages simpler local shading models. Along this axis, rasterization-based differentiable renderers [10, 27, 36, 25, 16] approximate gradients by generating derivatives from projected pixels to 3D parameters. These methods are restricted to primary visibility and ignore indirect lighting effects by construction, but their simplicity and efficiency offer an attractive trade-off for 3D reconstruction. We follow this line of work and build atop DIB-R [10, 22] by augmenting its shading models with physics-based ones.

**Learning-based Inverse Graphics.** Recent research on inverse graphics targets the ill-posed problem of jointly estimating geometry, reflectance and illumination from image observations using neural networks. For single image inverse rendering, one dominant approach is to employ 2D CNNs to learn data-driven features and use synthetic data as supervision [42, 49, 35, 33, 52], but these methods do not always generalize to complex real-world images [4]. To overcome the data issue, a recent body of work investigates the use of self-supervised learning to recover scene intrinsics [1], including domain adaptation from synthetic reflectance dataset [37], object symmetry [54, 53], or multi-illumination images depicting the same scene [30, 32, 39, 58]. However, these methods either rely on specific priors or require data sources tedious to capture in practice. Some works tackle the subtask of lighting estimation only [18, 17, 20], but still need to carefully utilize training data that are hard to capture. Most similar to us, DIB-R [10] tackles unsupervised inverse rendering in the context of differentiable rendering. Zhang et al. [62] further combines DIB-R with StyleGAN [24] generated images to extract and disentangle 3D knowledge. These works perform inverse rendering from real image collections without supervision, but may fail to capture complex material and lighting effects—in contrast, our method models these directly. Several techniques also try to handle more photorealistic effects but typically require complex capturing settings, such as controllable lighting [28, 29], a co-located camera-flashlight setup [41, 13, 34, 5, 6, 8, 48, 38], and densely captured multi-view images [14, 55, 7, 60] with additional known lighting [19] or hand-crafted inductive labels [43]. In our work, we propose a hybrid differentiable renderer and learn to disentangle complex specular effects given a single image. Similar to the recent NeRD [7] and PhySG [60] which recover non-Lambertian reflectance and illumination with a spherical Gaussian (SG) basis [51], we also employ SGs to model the SV-BRDF and incident lighting, but apply this representation to mesh-based differentiable rendering with direct access to the surface.

## 3   Differentiable Deferred Rendering

In this section, we introduce DIB-R++, our differentiable rendering framework based on deferred shading [12]. DIB-R++ is a hybrid differentiable renderer that can efficiently approximate direct illumination and synthesize high-quality images. Concretely, our renderer leverages the differentiable

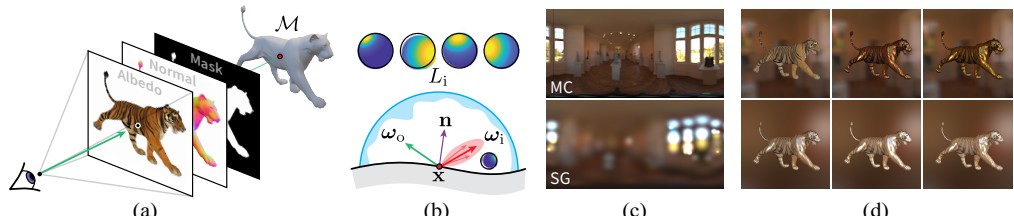

Figure 1: **Overview.** Given a 3D mesh $\mathcal{M}$, we employ (a) a rasterization-based renderer to obtain diffuse albedo, surface normals and mask maps. In the shading pass (b), we then use these buffers to compute the incident radiance by sampling or by representing lighting and the specular BRDF using a spherical Gaussian basis. Depending on the representation used in (c), we can recover a wide gamut of specular/glossy appearances (d).

rasterization framework of DIB-R [10] to recover shape attributes and further employs physics-based material and lighting models to estimate appearance.

## 3.1 Overview

We provide an overview of our technique in Fig. 1. We first rasterize a 3D mesh to obtain diffuse albedo and material maps, surface normals, and a silhouette mask. This information is deferred to the shading pass, where outgoing radiance is either estimated stochastically or approximated using a spherical Gaussian basis. The rasterizer and shader are differentiable by design, allowing gradients to be propagated to lighting, material and shape parameters for downstream learning tasks.

## 3.2 Background

Our goal is to provide a differentiable formulation of the rendering process to enable fast inverse rendering from 2D images. Let $\mathcal{M}$ be a 3D object in a virtual scene. We start from the (non-emissive) rendering equation (RE) [23], which states that the outgoing radiance $L_o$ at any surface point $\mathbf{x} \in \mathcal{M}$ in the camera direction $\boldsymbol{\omega}_o$ is given by

$$L_o(\mathbf{x}, \boldsymbol{\omega}_o) = \int_{\mathcal{H}^2} f_r(\mathbf{x}, \boldsymbol{\omega}_i, \boldsymbol{\omega}_o) L_i(\mathbf{x}, \boldsymbol{\omega}_i) |\mathbf{n} \cdot \boldsymbol{\omega}_i| \, d\boldsymbol{\omega}_i, \tag{1}$$

where $L_i$ is the incident radiance, $f_r$ is the (spatially-varying) bidirectional reflectance distribution function (SV-BRDF) and $\mathbf{n}$ is the surface normal at $\mathbf{x}$. The domain of integration is the unit hemisphere $\mathcal{H}^2$ of incoming light directions $\boldsymbol{\omega}_i$. The BRDF characterizes the surface's response to illumination from different directions and is modulated by the cosine foreshortening term $|\mathbf{n} \cdot \boldsymbol{\omega}|$. Intuitively, Eq. (1) captures an energy balance and computes how much light is received and scattered at a shading point in a particular direction.

Estimating the RE typically requires Monte Carlo (MC) integration [46], which involves tracing rays from the camera into the scene. Albeit physically correct, this process is computationally expensive and does not generally admit a closed-form solution. MC estimators can exhibit high variance and may produce noisy pixel gradients at low sample count, which may significantly impact performance and convergence. To keep the problem tractable, we thus make several approximations of Eq. (1), which we detail in the next section.

## 3.3 Two-stage Deferred Rendering

We now describe our rendering framework (Fig. 1). We start by defining three families of parameters, where $\boldsymbol{\pi} \in \mathbb{R}^{d_\pi}$ encodes the shape attributes (e.g., vertex positions), $\boldsymbol{\theta} \in \mathbb{R}^{d_\theta}$ describes the material properties, and $\boldsymbol{\gamma} \in \mathbb{R}^{d_\gamma}$ captures the illumination in the scene. In what follows, we shall only consider a single pixel, indexed by $p$, within an RGB image $I \in \mathbb{R}_+^{3 \times h \times w}$ for notational simplicity.

**Stage 1: Rasterization Pass.** We first employ a differentiable rasterizer $R$ [10] to generate primary rays $\boldsymbol{\omega}_o \in \mathcal{S}^2$ from a camera and render our scene $\mathcal{M}$ into geometry buffers (commonly called *G-buffers*) containing the surface intersection point $\mathbf{x}_p \in \mathbb{R}^3$, the surface normal $\mathbf{n}_p \in \mathcal{S}^2$, and the spatially-varying material parameters $\boldsymbol{\theta}_p$ (e.g., diffuse albedo). This rendering pass also returns a visibility mask $v_p \in \{0, 1\}$ indicating whether pixel $p$ is occupied by the rendered object, separating the foreground object $I_f$ from its background environment $I_b$ so that $I = I_f + I_b$. We have:

$$R(\mathcal{M}, p, \boldsymbol{\omega}_o) = (\mathbf{x}_p, \mathbf{n}_p, \boldsymbol{\theta}_p, v_p). \tag{2}$$

**Stage 2: Shading Pass.** Given surface properties and outgoing direction $\boldsymbol{\omega}_o$, we then approximate the outgoing radiance $L_o(\mathbf{x}_p, \boldsymbol{\omega}_o)$ through several key assumptions. First, we restrict ourselves to direct illumination only (i.e. single-bounce scattering) and assume that the incoming radiance is given by a distant environment map $L_i : \mathcal{S}^2 \to \mathbb{R}_+^3$. Therefore, we do not model self-occlusion and

$L_i(\mathbf{x}_p, \boldsymbol{\omega}_i) \equiv L_i(\boldsymbol{\omega}_i; \boldsymbol{\gamma})$. Such simplification largely reduces computation and memory costs and is trivially differentiable. Second, we assume that the material parameters $\boldsymbol{\theta}$ can model both diffuse and specular view-dependent effects. At a high level, we define our shading model $S$ so that:

$$S(\mathbf{x}_p, \mathbf{n}_p, \boldsymbol{\omega}_o; \boldsymbol{\theta}_p, \boldsymbol{\gamma}) \approx L_o(\mathbf{x}_p, \boldsymbol{\omega}_o). \tag{3}$$

Importantly, a differentiable parameterization of $S$ enables the computation of pixel gradients with respect to all scene parameters $\boldsymbol{\Theta} = (\boldsymbol{\pi}, \boldsymbol{\theta}, \boldsymbol{\gamma})$ by differentiating $I_p(\boldsymbol{\Theta}) = (S \circ R)(\mathcal{M}, p, \boldsymbol{\omega}_o)$. Given a scalar objective function defined on the rendered output $I$, $\partial I / \partial \boldsymbol{\pi}$ is computed using DIB-R [10]. In what follows, we thus mainly focus on formulating $\partial I / \partial \{\boldsymbol{\theta}, \boldsymbol{\gamma}\}$ so that all gradients can be computed using the chain rule, allowing for joint optimization of geometry, material and lighting parameters. We assume henceforth a fixed pixel $p$ for conciseness, and remove the subscript.

### 3.4 Shading Models

Since our primary goal is to capture a wide range of appearances, we provide two simple techniques to approximate Eq. (1): Monte Carlo (MC) and spherical Gaussians (SG). The former targets more mirror-like objects and can better approximate higher frequencies in the integrand, but is more expensive to compute. The latter is more robust to roughness variations but is limited by the number of basis elements. To model reflectance, we choose to use a simplified version of the isotropic Disney BRDF [9, 15] based on the Cook–Torrance model [11], which includes diffuse albedo $\mathbf{a} \in [0, 1]^3$, specular albedo $s \in [0, 1]$, surface roughness $\beta \in [0, 1]$ and metalness $m \in [0, 1]$. Metalness allows us to model both metals and plastics in a unified framework. We let the diffuse albedo vary spatially ($\mathbf{a} = \mathbf{a}(\mathbf{x})$) and *globally* define all other attributes to restrict the number of learnable parameters.

**Monte Carlo Shading.** Given a surface point $\mathbf{x} \in \mathcal{M}$ to shade, we importance sample the BRDF to obtain $N$ light directions $\boldsymbol{\omega}_i^k$ and compute the BRDF value. We represent the incident lighting $L_i^{(\mathrm{MC})}$ as a high-dynamic range image $\boldsymbol{\gamma} \in \mathbb{R}_+^{3 \times h_l \times w_l}$ using an equirectangular projection, which can be queried for any direction via interpolation between nearby pixels. The final pixel color is then computed as the average over all samples, divided by the probability of sampling $\boldsymbol{\omega}_i^k$:

$$S^{(\mathrm{MC})}(\mathbf{x}, \mathbf{n}, \boldsymbol{\omega}_o; \boldsymbol{\theta}, \boldsymbol{\gamma}) = \frac{1}{N} \sum_{k=1}^{N} \frac{f_r(\mathbf{x}, \boldsymbol{\omega}_i^k, \boldsymbol{\omega}_o; \boldsymbol{\theta}) \, L_i^{(\mathrm{MC})}(\boldsymbol{\omega}_i^k; \boldsymbol{\gamma}) \, |\mathbf{n} \cdot \boldsymbol{\omega}_i^k|}{p(\boldsymbol{\omega}_i^k)}. \tag{4}$$

When the surface is near-specular (e.g., a mirror), one can efficiently estimate the RE as reflected rays are concentrated in bundles (e.g., to satisfy the law of reflection). However, this estimator can suffer from high variance for rougher surfaces; a higher number of samples may be necessary to produce usable gradients. While this can be partially improved with multiple importance sampling [50], emitter sampling would add a significant overhead due to the environment map being updated at every optimization step. This motivates the use of a more compact representation.

**Spherical Gaussian Shading.** To further accelerate rendering while preserving expressivity in our shading model, we use a spherical Gaussian (SG) [51] representation. Projecting both the cosine-weighted BRDF and incident radiance into an SG basis allows for fast, analytic integration within our differentiable shader, at the cost of some high frequency features in the integrand. Concretely, an SG kernel has the form $\mathcal{G}(\boldsymbol{\omega}; \boldsymbol{\xi}, \lambda, \boldsymbol{\mu}) = \boldsymbol{\mu} \, e^{\lambda(\boldsymbol{\xi} \cdot \boldsymbol{\omega} - 1)}$, where $\boldsymbol{\omega} \in \mathcal{S}^2$ is the input spherical direction to evaluate, $\boldsymbol{\xi} \in \mathcal{S}^2$ is the axis, $\lambda \in \mathbb{R}_+$ is the sharpness, and $\boldsymbol{\mu} \in \mathbb{R}_+^3$ is the amplitude of the lobe. We represent our environment map using a mixture of $K$ lighting SGs $\mathcal{G}_l$, so that:

$$L_i^{(\mathrm{SG})}(\boldsymbol{\omega}_i; \boldsymbol{\gamma}) \approx \sum_{k=1}^{K} \mathcal{G}_l^k(\boldsymbol{\omega}_i; \boldsymbol{\xi}_l^k, \lambda_l^k, \boldsymbol{\mu}_l^k), \tag{5}$$

where $\boldsymbol{\gamma} := \{\boldsymbol{\xi}_l^k, \lambda_l^k, \boldsymbol{\mu}_l^k\}_k$. For the BRDF, we follow Wang et al. [51] and fit a single, monochromatic SG to the specular lobe so that $f_r^{(\mathrm{SG})}$ is a sum of diffuse and specular lobes. The full derivation can be found in our supplementary material (Sec. A). Finally, we approximate the cosine foreshortening term using a single SG $|\mathbf{n} \cdot \boldsymbol{\omega}_i| \approx \mathcal{G}_c(\boldsymbol{\omega}_i; \mathbf{n}, 2.133, 1.17)$ [40]. Regrouping all terms, the final pixel color can be computed as:

$$S^{(\mathrm{SG})}(\mathbf{x}, \mathbf{n}, \boldsymbol{\omega}_o; \boldsymbol{\theta}, \boldsymbol{\gamma}) = \int_{\mathcal{S}^2} f_r^{(\mathrm{SG})}(\mathbf{x}, \boldsymbol{\omega}_i^k, \boldsymbol{\omega}_o; \boldsymbol{\theta}) \, L_i^{(\mathrm{SG})}(\boldsymbol{\omega}_i; \boldsymbol{\gamma}) \, \mathcal{G}_c(\boldsymbol{\omega}_i) \, \mathrm{d}\boldsymbol{\omega}_i, \tag{6}$$

which has an analytic form that can be automatically differentiated inside our renderer. All parameters of the SGs, as well as the BRDF parameters, are learnable.

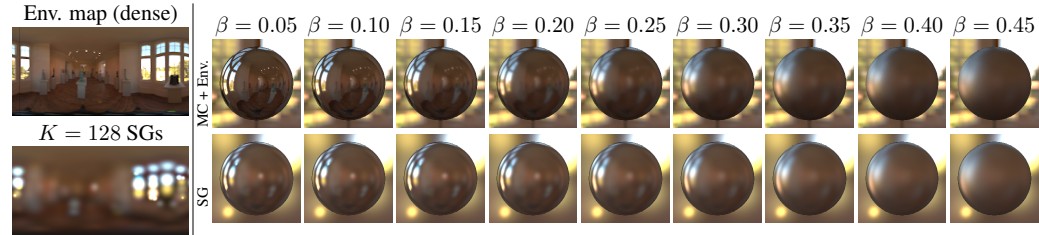

Figure 2: **Visual comparisons between MC and SG shading.** On the left, we show an environment map and its SG-representation ($K = 128$). While losing sharp details, SGs only needs 1% parameters compared to the dense pixel HDR map. On the right, we increase roughness left-to-right, revealing that our spherical basis can correctly approximate direct illumination at moderate-to-high roughness.

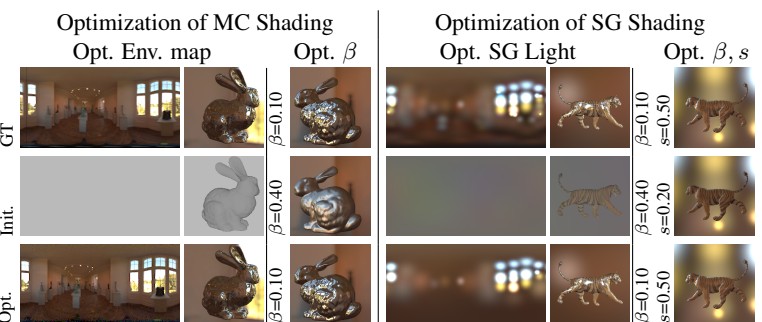

Figure 3: **Optimization**. We optimize complex lighting and material for both MC shading (left) and SG shading (right). In each case, we first optimize lighting and show the light and rendered image in the first block. Next we optimize the surface material and show the surface material value and rendered image in the second block.

**Comparison.** To visually compare our two shading techniques and understand their limitations, we render a unit sphere under the same lighting (represented differently) in Fig. 2. To do so, we first fit a HDR environment map with $K = 128$ SGs using an equirectangular projection. As shown on the left, SGs smooth out high frequency details and sharp corners but require much fewer parameters to reconstruct incident lighting (896 vs. 98 304). On the right, we visualize the effect of increasing the surface roughness $\beta$ under the corresponding light representation.

Intuitively, this point of diminishing return indicates that MC is only so useful when the surface reflects most of the incoming light (e.g., a mirror). Indeed, when $\beta$ is small enough ($\beta \to 0$) and we deal with a highly non-Lambertian surface, a small number of MC samples are enough to estimate direct illumination, which in turn implies faster render speed and low memory cost. On the other end of the spectrum ($\beta \to 1$), significantly more samples (e.g., $N > 1000$) are needed to accurately integrate incident light, resulting in longer inference times. In such a case, SGs should be favored since they offer a significant improvement. In the absence of any prior knowledge on the material type, SG shading is preferred. This is reflected in our experiments in Sec. 5-6.

**Optimization.** We perform a sanity check on our renderer in Fig. 3 by optimizing for lighting and reflectance properties from a multi-view image $L_1$-loss with fixed geometry. We show the ground-truth (GT) parameters and rendered images in the first row, along with initial parameters (Init.) in the second row. Here, we optimize parameters separately using gradient-descent while the others are kept fixed to validate each component of our shading model. DIB-R++ can successfully estimate material and lighting parameters, including the environmental lighting, surface roughness and specular albedo of the object. We find that the converged material parameters closely match GT, while the optimized environment map loses some details due to gradients coming entirely from surface reflections (foreground supervision only). In particular, surface highlights are well captured by our technique. For more optimization results, please refer to the supplementary (Sec. C).

## 4 Application: Single Image 3D Reconstruction

We demonstrate the effectiveness of our hybrid framework through the learning-based problem of single image 3D reconstruction without supervision. While previous works [10, 62] generally focus on diffuse illumination only, our goal is to jointly infer geometry, reflectance, and lighting from a single image $\tilde{I}$ containing strong specular transport. To this end, we employ a convolutional neural network $F$, parameterized by learnable weights $\vartheta$, to predict 3D attributes of a mesh $\mathcal{M}$ with pre-determined topology (sphere in our case). We adopt the U-Net [47] architecture of the original DIB-R [10, 62] and modify its output to also predict the appropriate BRDF attributes $\theta$ and light parameters $\gamma$ (pixel colors or SG coefficients) so that $F(\tilde{I}; \vartheta) = (\pi, \theta, \gamma)$. We then render these parameters back to an image $I$ using our differentiable renderer and apply a loss $\mathcal{L}$ on the RGB output

to compare the input image $\tilde{I}$ and the rendered image $I$, where:

$$\mathcal{L}(\boldsymbol{\vartheta}) = \alpha_{\text{im}}\mathcal{L}_{\text{im}}(\tilde{I}, I) + \alpha_{\text{msk}}\mathcal{L}_{\text{msk}}(\tilde{V}, V) + \alpha_{\text{per}}\mathcal{L}_{\text{per}}(\tilde{I}, I) + \alpha_{\text{lap}}\mathcal{L}_{\text{lap}}(\boldsymbol{\pi}). \tag{7}$$

Similar to DIB-R [10], we combine multiple consistency losses with regularization terms: $\mathcal{L}_{\text{im}}$ is an image loss computing the $L^1$-distance between the rendered image and the input image, $\mathcal{L}_{\text{msk}}$ is an Intersection-over-Union (IoU) loss of the rendered silhouette $V$ and the input mask $\tilde{V}$ of the object [25], $\mathcal{L}_{\text{per}}$ is a perceptual loss [21, 61] computing the $L^1$-distance between the pre-trained AlexNet [26] feature maps of rendered image and input image, and $\mathcal{L}_{\text{lap}}$ is a Laplacian loss [36, 25] to penalize the change in relative positions of neighboring vertices. We set $\alpha_{\text{im}} = 20$, $\alpha_{\text{msk}} = 5$, $\alpha_{\text{per}} = 0.5$, $\alpha_{\text{lap}} = 5$, which we empirically found worked best.

# 5 Evaluation on Synthetic Datasets

We conduct extensive experiments to evaluate the performance of DIB-R++. We first quantitatively evaluate on synthetic data where we have access to ground-truth geometry, material and lighting. Since MC ad SG shading have individual pros and cons, we validate them under different settings. In particular, we generate separate datasets with two different surface materials: purely **metallic** surfaces with no roughness, and **glossy** surfaces with random positive roughness. We compare the performance of both shading models against the baseline method [10].

**Synthetic Datasets.** We chose 485 different car models from *TurboSquid*[2] to prepare data for metallic and glossy surfaces. We also collected 438 freely available high-dynamic range (HDR) environment maps from *HDRI Haven*[3] to use as reference lighting, which contain a wide variety of illumination configurations for both indoor and outdoor scenes. To render all 3D models, we use Blender's Cycles[4] path tracer with the Principled BRDF model [9]. We create two datasets, Metallic-Surfaces and Glossy-Surfaces. For metallic surfaces, we set $\beta = 0$ and $m = 1$. Conversely, we set $m = 0$, $s = 1$ and randomly pick $\beta \in [0, 0.4]$ to generate images for glossy surfaces.

**Baseline.** We compare our method with the rasterization-based baseline DIB-R [10], which supports spherical harmonics (SH) lighting. While the original lighting implementation in [10] is monochromatic, we extend it to RGB for a fairer comparison. For quantitative evaluation, we first report the common $L^1$ pixel loss between the re-rendered image using our predictions and ground-truth (GT) image($L = \|\tilde{I} - I\|_1$), and 2D IoU loss between rendered silhouettes and ground-truth masks($L = 1 - \frac{\tilde{V} \odot V}{\tilde{V} + V}$). We experimentally find that these numbers are very close in different methods. Thus, we further evaluate the quality of diffuse albedo and lighting predictions using normalized cross correlation (NCC, $L = 1 - \frac{\sum \tilde{\gamma} \odot \gamma_{\text{pred}}}{\|\tilde{\gamma}\|_2 \|\gamma_{\text{pred}}\|_2}$, where $\gamma_{\text{pred}}$ is the predicted albedo and light while $\tilde{\gamma}$ is GT). We provide more details of these metrics in the supplementary material (Sec. E).

## 5.1 Metallic Surfaces

**Experimental Settings.** We first apply all methods to the metallic car dataset. Since this surface property is known a priori, we relax the task for MC shading by setting $\beta = 0$ and only predict geometry, diffuse albedo and lighting from the input image. This allows us to render MC at a low sample count ($N = 4$), achieving higher rendering speed and a lower memory cost. In particular, we predict the relative offset for all $|\mathcal{M}| = 642$ vertices in a mesh and a $256 \times 256$

| Shading | Metallic surfaces (↓) | | | | Glossy surfaces (↓) | | | |
|---|---|---|---|---|---|---|---|---|
| | Image | 2D IoU | Light | Tex. | Image | 2D IoU | Light | Tex. |
| MC | 0.019 | 0.062 | **0.074** | **0.152** | 0.024 | 0.061 | 0.106 | 0.142 |
| SG | 0.019 | 0.069 | 0.095 | 0.218 | 0.024 | 0.057 | **0.091** | **0.140** |
| SH [10] | 0.019 | 0.056 | 0.220 | 0.206 | 0.024 | 0.062 | 0.131 | 0.152 |

Table 1: Quantitative results of single image 3D Reconstruction on synthetic data. While all the methods achieve comparable performance on re-rendered images and 2D IoUs, both MC and SG achieve better results on lighting and texture. MC is particularly better for metallic surfaces, and SG works best for glossy surfaces.

texture map, following the choices in [10]. We also predict a $256 \times 256$ RGB environment map. For SG shading, we predict all parameters. While shape and texture are the same as MC shading, we adopt $K = 32$ for SG and predict two global parameters $\beta$ and $s$ for the specular BRDF. This keeps the number of parameters relatively low while providing enough flexibility to capture different radiometric configurations.

---

[2] https://turbosquid.com. We obtain consent via agreement with TurboSquid, following their license at https://blog.turbosquid.com/turbosquid-3d-model-license.

[3] https://hdrihaven.com. We follow the CCO license at https://hdrihaven.com/p/license.php.

[4] https://blender.org

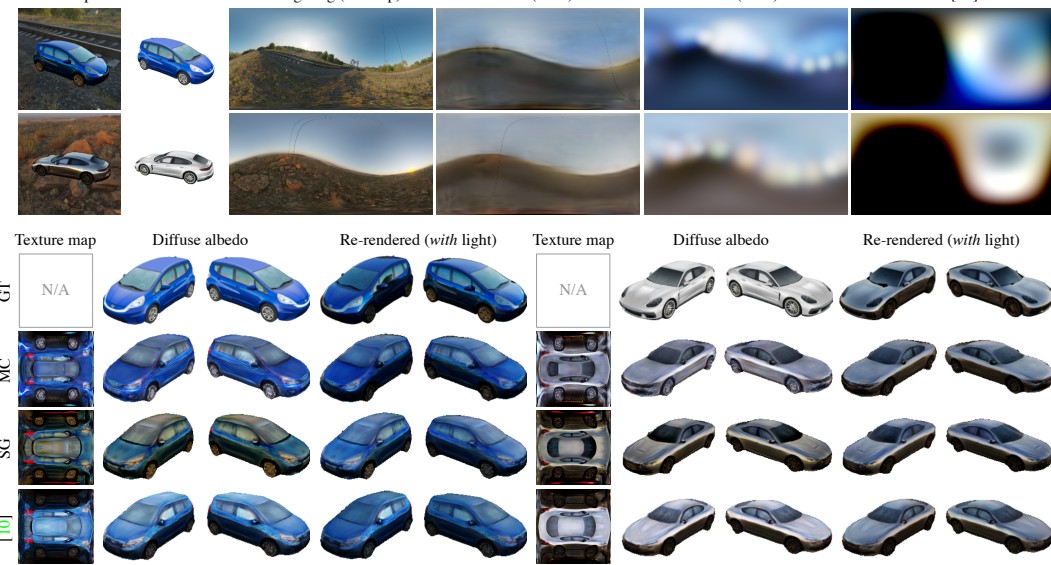

Figure 4: **Prediction results on the metallic car dataset** ($\beta = 0$). While the re-rendered images look similar, the underlying components (material and lighting) are different. A SH basis [10] cannot recover the high frequency details of the sky light maps. In this case, MC performs best due to low variance in the estimator for mirror-like BRDFs. SGs can recover the overall form and contrast of the light map, but tend to predict incorrect texture maps incorporating the ground dominant color (e.g., brown). Note that GT texture maps are not available as they cannot be compared due to different $uv$-parameterizations / texture atlases.

**Experimental Results.** Quantitative and qualitative results are shown in Fig. 4 and Table 1 (Left), respectively. Since the main loss function comes from the difference between GT and re-rendered images, we find the re-rendered images (with light) from the predictions are all close to the GT image for different methods, and quantitatively, the image loss and 2D IoU loss are also similar across different models. However, we observe significant differences on the predicted albedo and lighting. Specifically, in Fig. 4, MC shading successfully predicts cleaner diffuse albedo maps and more accurate lighting, while Chen et al. [10] "bakes in" high specular effects into the texture. Quantitatively, we outperform [10] with a $3\times$ improvement in terms of NCC loss for lighting, demonstrating the effectiveness of our DIB-R++. We further compare MC shading with SG shading. While SG shading achieves reasonable lighting predictions, it fails to reconstruct the high frequency details in the lighting and has circular spot effects caused by the isotropic SGs. Finally, we note that due to the ambiguity of the learning task, the overall intensities of all predicted texture maps can largely vary. Still, we observe that MC can better recover fine details, such as the wheels' rims.

### 5.2 Glossy Surfaces

**Experimental Settings.** We further apply our model to synthetic images rendered with positive roughness (glossy). To apply MC shading in such a case, we assume no prior knowledge for material and use a high sample count ($N = 1024$) to account for possibly low roughness images in the dataset. Due to high rendering time and memory cost, we subsample $4\%$ of the pixels and apply $\mathcal{L}_{\text{im}}$ to those pixels only in each training iteration. As such, we do not use the perceptual loss $\mathcal{L}_{\text{per}}$ as it relies on the whole image. We predict a $32 \times 16$ environment map for lighting and predict global $\beta$ and $m$ for the specular BRDF. For SG shading, we use the same settings as for the metallic surfaces.

**Experimental Results.** We apply both MC and SG shading and compare with [10]. Results are shown in Fig. 5 and Table 1 (Right). Qualitatively, SG shading has better lighting predictions with correct high-luminance regions. The specular highlights in the images successfully guide SG shading, while the bright reflection on the car window and front cover are fused with texture map in [10]. MC also has reasonable lighting predictions, but the predicted light map lacks structure due to weak surface reflections. Without a perceptual loss term, the predicted textures also tend to be blurrier.

Quantitatively, SG shading significantly outperforms [10] on lighting prediction in terms of NCC (0.078 vs. 0.127) and improves on texture prediction in terms of BRDF/lighting disentanglement. We also compare with MC shading, where MC achieves slightly worse results on lighting predictions compared to SG, but is still much better than [10].

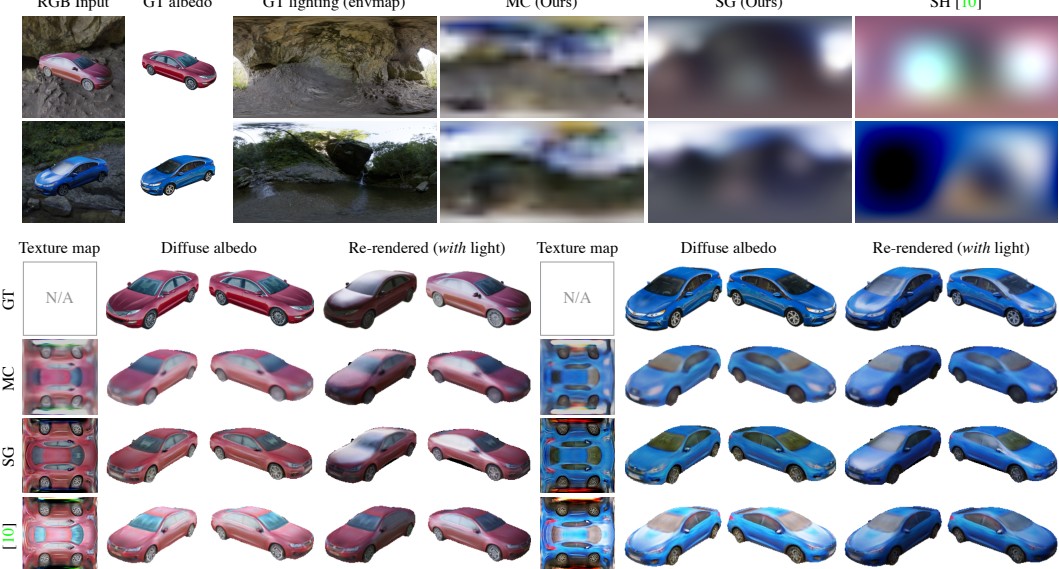

Figure 5: **Prediction results on the glossy car dataset** ($\beta > 0$). When the objects are glossy but not perfectly specular, SGs can correctly disentangle reflectance from lighting, as evidenced by the absence of white highlights in the predict albedos. Chen et al. [10] cannot capture these bright regions due to a diffuse-only shading model, while MC oversmooths the predictions due to noisier pixel gradients. While all methods cannot completely reconstruct the environment map, our method can predict the correct dominant light location and sky color.

## 5.3 Discussion

As shown in our previous two experiments, Monte Carlo shading works best under a metallic assumption ($\beta = 0$), in which case the rendered images can have rich details at low sample count ($N \leq 4$). However, when the surface is more Lambertian (i.e., when $\beta$ is becoming larger), we have to compensate with a larger $N$ to produce noise-free renderings, which impacts learning both time- and memory-wise. As a consequence, we recommend applying MC shading to metallic surfaces only, and default to SGs otherwise.

Our spherical Gaussian shading pipeline provides an analytic formulation for estimating the rendering equation, which avoids the need of tracing ray samples, largely accelerating the rendering process. While SGs can be blurry on metallic surfaces, in most case (e.g., when $\beta \geq 0.2$) it can model similar rendering effects at a fraction of the cost, achieving better results than MC shading and [10].

After inspecting the predicted surface material properties ($\beta$, $s$, $m$) and diffuse albedo with the ground-truth parameters in Blender, we find the materials contain little correlation and the intensities of diffuse albedo might change. As for SG, we are using only 32 basis elements to simulate a complex, high definition environment map (2K). Since SGs can only represent a finite amount of details, we find the predicted global $\beta$ tends to be too small. One hypothesis for this is that the optimizer artificially prefers more reflections (and thus lower roughness) to be able to estimate at least some portions of the environment map. On the other hand, in MC, due to the absence of a perceptual loss, the predicted texture is too blurry and cannot represent GT to a high detail. We find that the predicted $\beta$ and $m$ do not have strong correlation with the GT material. Lastly, we note that the predicted texture map has to change its overall intensity to accommodate for other parameters to ensure the re-rendered images are correct, which leads to some differences with GT. More analysis can be found in supplementary (Sec. C and F).

In summary, when we have no prior knowledge about the material, our re-rendered images can be very close to the input images but the predicted material parameters are not always aligned with the GT materials. We believe this problem can be relieved by incorporating additional local constraints, e.g., part-based material priors, or by leveraging anisotropic SGs [56]. For instance, a car body is metallic while its wheels are typically diffuse; predicting different parameters for each region has the potential of improving disentanglement and interpretability. We leave this as future work.

## 6 Evaluation on a Real-world Dataset

We further qualitatively evaluate our method via training on StyleGAN generated data and testing on real imagery, following the pipeline in [62] in Sec. 6.1, and use our predictions to perform artistic manipulation in Sec. 6.2.

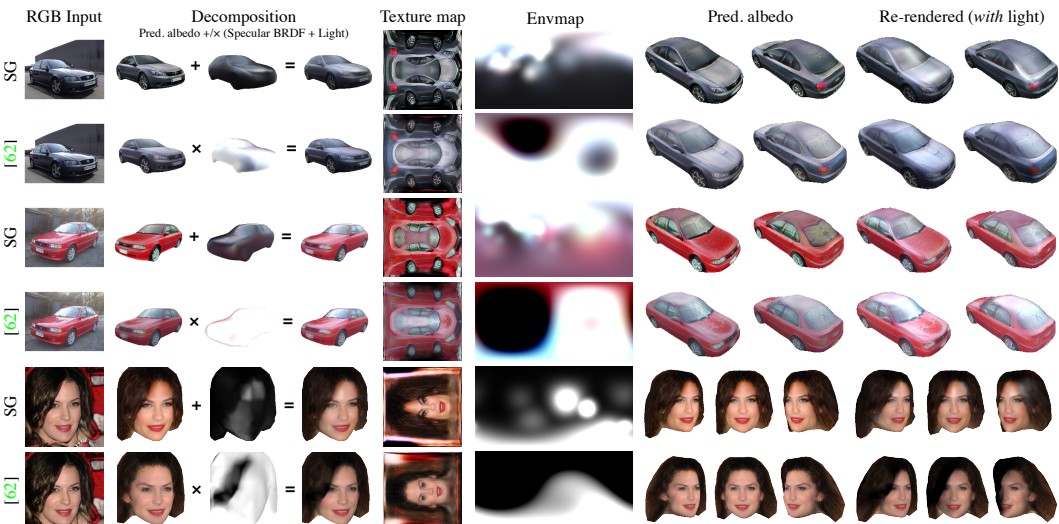

Figure 6: **Results on real imagery from the StyleGAN-generated dataset (cars and white female faces).** Our method can recover a meaningful decomposition as opposed to [62], as shown by cleaner texture maps and directional highlights (e.g., car windshield). Even when using monochromatic lighting on faces, our method can correctly predict the specular highlights on the forehead and none in the hair, while SH produces dark artifacts.

## 6.1 Realistic Imagery

**Experimental Settings.** Our DIB-R++ can also be applied to learn 3D properties from realistic imagery. Following [62], we use StyleGAN [24] to generate multi-view images of cars and faces, which is the data we need to train our model. The generated objects contains cars under various lighting conditions, ranging from high specular paint to nearly diffuse. Thus, we only apply SG shading and adopt the same setting, where we predict $|\mathcal{M}| = 642$ vertex movements, a $256 \times 256$ diffuse texture map, $K = 32$ SG bases and two global $\beta$ and $s$. We also compare [62] as the baseline on the same dataset by using the same training procedure.

**Experimental Results on StyleGAN Dataset.** In the absence of ground-truth on the StyleGAN generated data, we qualitatively evaluate our results and compare with [62] in Fig. 6. Our DIB-R++ reconstructs more faithful material and lighting components, producing an interpretable decomposition. Specifically, our model can represent the dominant light direction more accurately, while naive shading tend to merge reflectance with lighting. We also provide an example with monochromatic lighting on a face example to reduce the degrees of freedom for the SH representation, yet it cannot correctly model light.

**Extension to Real Imagery from LSUN Dataset.** Our model is trained on synthetic data [62] generated by StyleGAN [24]. Thanks to this powerful generative model, the distribution of GAN images is similar to the distribution of real images, allowing our model to generalize well. We show reconstruction results on real images from LSUN [57] in Fig. 7. We provide more results in the supplementary (Sec. E), we also provide additional turntable videos on our project webpage.

During inference, we do not need any camera pose and predict shapes in canonical view. However, camera poses are needed to re-render the shape. Since ground-truth camera poses are not available for real images, we manually adjust the camera poses in Fig. 7. As a result, the re-rendered images are slightly misaligned with GT. However, DIB-R++ still accounts for specularities and predicts correct predominant lighting directions and clean textures.

## 6.2 Material Editing and Relighting

Finally, we demonstrate some applications of DIB-R++ to artistic manipulation in Fig. 8. On the left, we show examples of editing the diffuse albedo, where we can insert text, decals or modify the base tint. Since our textures are not contaminated by lighting, clean texture maps can be easily edited by hand and the re-rendered images look natural. On the right, we show examples of editing lighting and surface materials, where we rotate the light (top) or increase glossiness (bottom). We also showcase results where we change lighting orientation or modify the object's glossiness with consistent shading, which is not feasible with a naive, Lambertian-only shading model.

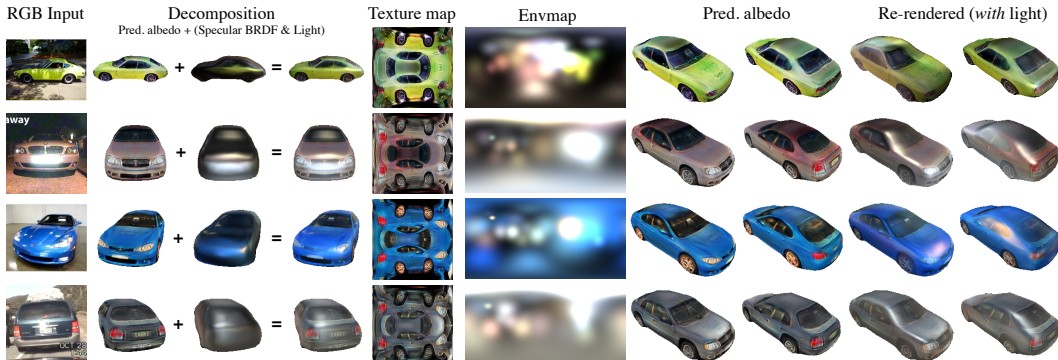

Figure 7: **Prediction on LSUN Dataset (Cars).** DIB-R++, trained on StyleGAN dataset, can generalize well to real images. Moreover, it also predicts correct high specular lighting directions and usable, clean textures.

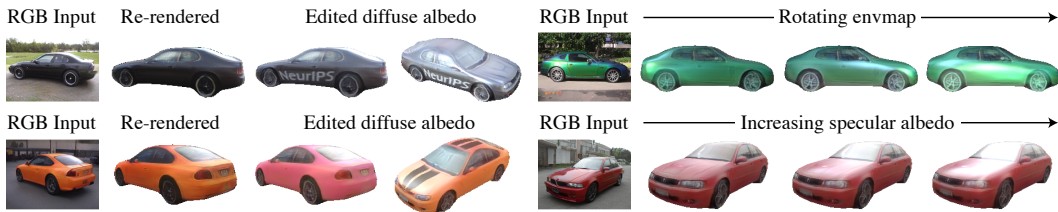

Figure 8: Our SG method allows for artistic manipulation of appearance, such as novel view synthesis, material editing (both diffuse and specular components), and relighting, thanks to our effective disentanglement.

# 7  Conclusion

We presented DIB-R++, a hybrid differentiable renderer that can effectively disentangle material and lighting. When embedded in a learning framework for single image 3D reconstruction, our method produces state-of-the-art results, and enables applications such as material editing and relighting. One limitation of our method is that the predicted base color may sometimes "bleed" into the lighting predictions. Combining our technique with segmentation methods like DatasetGAN [63] could alleviate this issue for a more practical, artist-friendly disentanglement. Moreover, the predicted reflections are sometimes blurrier than ground-truth; this is mainly due to a limited number SG components for lighting and could potentially be improved with a larger mixture. Finally, on some occasions, the diffuse albedo in our synthetic dataset have baked-in reflections (e.g., GT red car in Fig. 5) instead of a uniform base color, which obfuscates the learning process. This could be mitigated by using more advanced physics-based materials such as those modeling clear coats.

**Broader Impact.** Our work focuses on disentangling geometry from appearance using a differentiable renderer, a relatively nascent research area. We show that augmenting a rasterization-based renderer with physics-based shading models improves reconstruction and allows for easier integration within larger machine learning pipelines. DIB-R++ relies on simple topology and strong data prior assumptions to produce useful decompositions; therefore it cannot generalize to the complexity and multi-modality of real-world scenes in its current form. Nonetheless, we believe that our work takes an important step in the joint estimation of shape, material and environmental lighting from a single image and we hope that it can advance applications in performance-oriented settings such as AR/VR, simulation technology and robotics. For instance, autonomous vehicles need to correctly assess their surroundings from limited signals; directly modeling light-surface interactions (e.g., specular highlights) may provide important cues to this end.

Like any ML model, DIB-R++ is prone to biases imparted through training data which requires an abundance of caution when applied to sensitive applications. For example, it needs to be carefully inspected when it is used to recover the 3D parameters of human faces and bodies as it is not tailored for them. It is not recommended in off-the-shelf settings where privacy or erroneous recognition can lead to potential misuse or any harmful application. For purposes of real deployment, one would need to carefully inspect and de-bias the dataset to depict the target distribution of a wide range of possible lighting conditions, skin tones, or at the intersection of race and gender.

**Disclosure of Funding.**   This work was funded by NVIDIA. Wenzheng Chen, Jun Gao and Zian Wang acknowledge additional indirect revenue in the form of student scholarships from University of Toronto and the Vector Institute. Joey Litalien acknowledges indirect funding from McGill University and the Natural Sciences and Engineering Research Council of Canada (NSERC).

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
