# Supplementary Material for
# DIB-R++: Learning to Predict Lighting and Material with a Hybrid Differentiable Renderer

**Wenzheng Chen**[1,2,3]  **Joey Litalien**[4,*]  **Jun Gao**[1,2,3]  **Zian Wang**[1,2,3]

**Clement Fuji Tsang**[1]  **Sameh Khamis**[1]  **Or Litany**[1]  **Sanja Fidler**[1,2,3]

NVIDIA[1]  University of Toronto[2]  Vector Institute[3]  McGill University[4]

{wenzchen, jung, zianw, cfujitsang, skhamis, olitany, sfidler}@nvidia.com, joey.litalien@mail.mcgill.ca

In this supplementary, we first provide more details of BRDF model of DIB-R++ in Sec. A and rendering specifications in Sec. B. We then show an additional analysis on the joint optimization of lighting and material using our DIB-R++ in Sec. C. We provide the experimental details on our datasets in Sec. D, with more exeprimental results in Sec. E and ablation study in Sec. F.

## A  BRDF Model

For the BRDF, we use a simplified version of the isotropic Disney BRDF [1]:

$$f_{\text{r}}(\mathbf{x}, \boldsymbol{\omega}_{\text{i}}, \boldsymbol{\omega}_{\text{o}}) = \frac{\mathbf{a}}{\pi} + \frac{D(\boldsymbol{\omega}_{\text{h}})F(\boldsymbol{\omega}_{\text{o}}, \boldsymbol{\omega}_{\text{h}})G(\boldsymbol{\omega}_{\text{i}}, \boldsymbol{\omega}_{\text{o}})}{4|\mathbf{n} \cdot \boldsymbol{\omega}_{\text{i}}||\mathbf{n} \cdot \boldsymbol{\omega}_{\text{o}}|}, \tag{1}$$

where $\mathbf{a}$ is the diffuse albedo, $D$ is the (micro-)normal distribution function (NDF), $F$ is the Fresnel term, $G$ is a geometry or shadowing factor, and $\boldsymbol{\omega}_{\text{h}} = (\boldsymbol{\omega}_{\text{i}} + \boldsymbol{\omega}_{\text{o}})/\|\boldsymbol{\omega}_{\text{i}} + \boldsymbol{\omega}_{\text{o}}\|$ is the half-vector.

**MC Shading.**  For the NDF, we use the GGX/Trowbridge–Reitz distribution parameterized by a roughness parameter $\beta > 0$, which we map to $\alpha = (\beta + 1)^2/8$ to allow for more perceptually linear change [3]:

$$D^{(\text{MC})}(\boldsymbol{\omega}_{\text{h}}; \alpha) = \frac{\alpha^2}{\pi\big((\mathbf{n} \cdot \boldsymbol{\omega}_{\text{h}})^2(\alpha^2 - 1) + 1\big)^2}. \tag{2}$$

We importance sample this term directly using inverse transform sampling [10, 1]. For the geometry term, we use the Schlick model [11], which is given by

$$G(\boldsymbol{\omega}_{\text{i}}, \boldsymbol{\omega}_{\text{o}}) = G_1(\boldsymbol{\omega}_{\text{i}})G_1(\boldsymbol{\omega}_{\text{o}}), \qquad G_1(\boldsymbol{\omega}; \mathbf{n}, \alpha) = \frac{\mathbf{n} \cdot \boldsymbol{\omega}}{(\mathbf{n} \cdot \boldsymbol{\omega})(1 - \alpha) + \alpha}. \tag{3}$$

Finally, for the Fresnel reflection coefficient, we use the SG approximation[2] [3] given by:

$$F(\boldsymbol{\omega}_{\text{o}}, \boldsymbol{\omega}_{\text{h}}) = s + (1 - s)2^{(-5.55473(\boldsymbol{\omega}_{\text{o}} \cdot \boldsymbol{\omega}_{\text{h}}) - 6.98316)(\boldsymbol{\omega}_{\text{o}} \cdot \boldsymbol{\omega}_{\text{h}})}, \tag{4}$$

where $s$ is the specular albedo at normal incidence, which we define as a linear combination between relative IORs and albedo using a *metallic* parameter $m \in [0, 1]$:

$$F_0 = \text{Lerp}\big(|1 - \eta|^2/|1 + \eta|^2, \mathbf{a}, m\big). \tag{5}$$

---

[*]Work done during an internship at NVIDIA.

[2]Note that this is not related to our SG model. It is common to replace the $1 - (\boldsymbol{\omega}_{\text{o}} \cdot \boldsymbol{\omega}_{\text{h}})^5$ term in Schlick's model by an SG to remove the power and improve efficiency.

35th Conference on Neural Information Processing Systems (NeurIPS 2021).

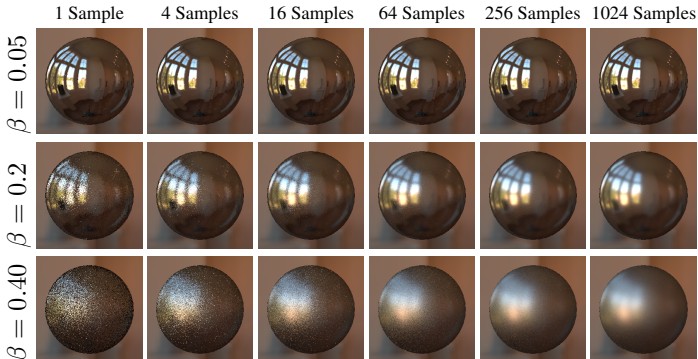

Figure A: **Sample Count vs. Roughness.** High roughness surfaces require more samples to render into noise-free images.

Here, $\eta = 1.5$ (vector) and $m$ is the blending weight. This interpolation allows us to treat conductors and dielectric with the same approximation. Intuitively, if a material is dielectric we use the IOR, otherwise we use the albedo to "tint" the reflection.

**SG Shading.** Similar to Wang et al. [12], we approximate the NDF $D$ using a single SG lobe

$$D^{(\text{SG})}(\boldsymbol{\omega}_{\text{h}}; \alpha) \approx \mathcal{G}_d\left(\boldsymbol{\omega}_{\text{h}}; \mathbf{n}, \frac{2}{\alpha^2}, \frac{1}{\pi\alpha^2}\right), \tag{6}$$

and apply a spherical warp [12] to orient the distribution lobe about the reflected view direction:

$$D^{(\text{SG Warped})}(\boldsymbol{\omega}_{\text{h}}; \alpha) \approx \mathcal{G}_d\left(\boldsymbol{\omega}_{\text{h}}; 2(\boldsymbol{\omega}_{\text{o}} \cdot \mathbf{n})\mathbf{n} - \boldsymbol{\omega}_{\text{o}}, \frac{\lambda_d}{4|\mathbf{n} \cdot \boldsymbol{\omega}_{\text{o}}|}, \boldsymbol{\mu}_d\right), \tag{7}$$

where $\lambda_d$ and $\boldsymbol{\mu}_d$ are those of Eq. (6). Since the Fresnel and geometry terms cannot be approximated with SGs, we assume that their values are constant across the entire BRDF lobe and pull them out of the integral. The cosine foreshortening term is approximated with a single SG as $|\mathbf{n} \cdot \boldsymbol{\omega}_{\text{i}}| \approx \mathcal{G}_c(\boldsymbol{\omega}_{\text{i}}; \mathbf{n}, 2.133, 1.17)$ [7]. The outgoing radiance can finally be evaluated in closed form by integrating against the environment map, also given as a mixture of SGs.

## B  Analysis of Rendering Effects

In this section, we analyze the rendering effects under different hyperparameters, including sampling count for MC shading and number of SG light components in SG shading. We also compare the rendering speed of MC shading and SG shading.

**Sampling Count.** As shown in Fig. A, we render a unit sphere with different roughness and sample count. When $\beta$ is close to 0, a small sample count (e.g. $N = 4$ samples) can render realistic images. However, as $\beta$ increases, more samples are needed to reduce noise. When $\beta$ is 0.2, we need $N = 256$ samples. When $\beta$ is 0.4, even $N = 1024$ samples may still generate noisy results.

**Number of SG Light Component.** As shown in Fig. B, we approximate the same environment map with different numbers of SG components in the mixture. Clearly, the more components we use, the more details we can represent. When we use 128 SGs ($128 \times 7 = 896$ parameters), we can approximate most details of the environment map with only $1\%$ parameters of the whole image ($256 \times 128 \times 3 = 98\,304$ parameters).

| Shading | Sample Count ($N$) | | | | | |
|---|---|---|---|---|---|---|
| MC | 1 | 4 | 16 | 64 | 256 | 1204 |
| Time (ms) | 4.0 | 3.7 | 5.0 | 9.1 | 28.3 | 292.5 |

| Shading | Number of SG Component ($K$) | | | | | | |
|---|---|---|---|---|---|---|---|
| SG | 1 | 2 | 4 | 8 | 16 | 32 | 64 | 128 |
| Time (ms) | 2.6 | 2.8 | 2.6 | 2.6 | 2.9 | 2.9 | 2.5 | 2.7 |

Table A: **Rendering Time Comparison.** We render a sphere into $256 \times 256$ images and compare the rendering time under different settings. In MC shading, more sample count costs higher rendering time, while in SG shading, a different numbers of SG component barely impacts timings.

**Rendering Speed.** We further evaluate the rendering performance of different shading methods. The results are shown in Tab. A. We implement both two shading methods in PyTorch [9]. Here, we only

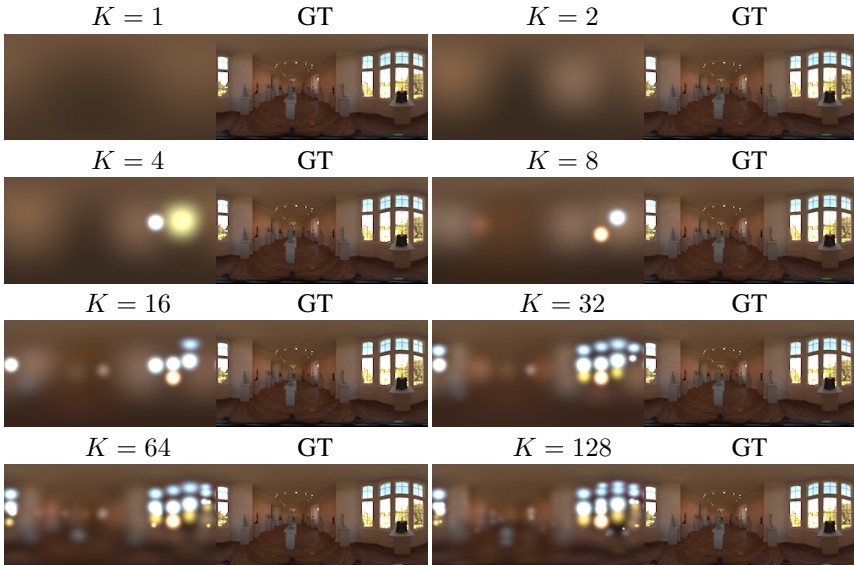

| $K = 1$ | GT | $K = 2$ | GT |
| $K = 4$ | GT | $K = 8$ | GT |
| $K = 16$ | GT | $K = 32$ | GT |
| $K = 64$ | GT | $K = 128$ | GT |

Figure B: **Number of SG Light Component**. More components (larger $K$) result in better environment map approximation.

consider the time cost for shading, without counting the cost of rasterization. For MC shading, time increases as sample count increases. For example, when the sample count changes from 256 to 1024, the rendering time becomes almost $10\times$ slower. Conversely, SG shading cost is nearly constant. One possible explanation for this is that the number of parameters of SG light is small (no more than 1000 parameters even for 128 SGs), and thus leads to almost constant time cost when computed in parallel with PyTorch. However, as more SGs can significantly impact memory cost, we choose $K = 32$ in the learning tasks.

**Comparison with Mitsuba 2 [8].** We further compare the rendering performance with Mitsuba2 [8] in Tbl. B. Since both our method and Mitusba 2 [8] supports ray tracing, we evaluate the running times and memory for the same ray tracing optimization task. We restrict the ray samples to be the same (4 samples) and run 300 iterations to optimize an environment map while fixing everything else. Our method performs faster (5.4sec v.s. 9.8sec) and requires less memory(1475MB v.s. 3287MB), which demonstrates DIB-R++ is highly performant.

|  | DIB-R++ | Mitsuba 2 [8] |
|---|---|---|
| Memory (MB) | 1475 | 3287 |
| Time (s) | 5.4 | 9.8 |

Table B: **Comparison with Mitsuba2 [8].** We evaluate the running time and memory within ray tracing optimization task. Our method performs faster and requires less memory compared to Mitusba2 [8].

## C   Joint Optimization of Lighting and Material

The design of our differentiable renderer allows for optimization over geometry, lighting and material parameters. To validate our technique, we showcase several examples of optimization in Fig. 3 in the main paper, where we optimize only one variable and keep the rest the same as GT. In this section, we further study joint optimization with more variables, a more challenging ill-posed problem. With advanced shading models, our DIB-R++ has the potential to recover complex photorealistic effects. We explore the extent to which our DIB-R++ can separate lighting and material (including diffuse albedo and surface material such as roughness and specular albedo) from only the rendered images, as shown in Fig. D and Fig. E.

### C.1   Disentanglement Analysis

Inverse rendering aims to recover shape, light and material from rendered images, where these scene properties are entangled during the complex image formation process. Thus, this problem is ill-posed as there might be several solutions for the same input image. As shown in Fig. C, we can get the same rendering with sharp lighting and large roughness, or smooth lighting and small roughness. Such ambiguities occur regardless of MC and SG shading.

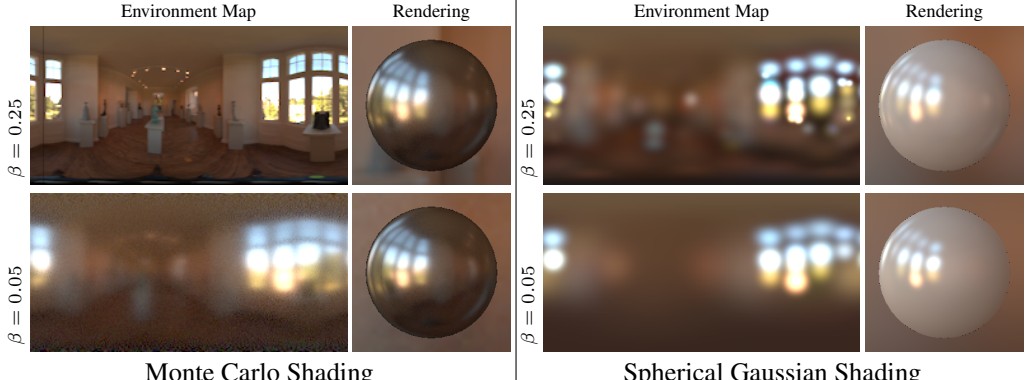

Figure C: **Disentanglement Analysis**. We render a sphere with 1) large roughness and sharp light (top row) and 2) small roughness and smooth light (bottom row), but their renderings are almost the same. It shows the ambiguity between light and surface material.

## C.2 Joint Optimization

To analyze the disentanglement behaviour, we render a unit sphere into 24 views, fix the shape and jointly optimize light and surface material from 2D images. We show Monte Carlo-based shading optimization in Fig. D and Spherical Gaussian-based shading optimization in Fig. E.

**MC Shading.** Several interesting properties can be found in Fig. D. First, DIB-R++ successfully optimizes the light and surface material such that the rendered images (Col. 4 & 7) are close to the GT images (Col. 1) after convergence. Moreover, the converged parameters are also reasonable, e.g. the optimized light recovers most high frequency details, especially in small roughness cases. Second, the converged parameters are not exactly the same as GT settings. For instance, compared to GT, the environment map is always brighter while the diffuse albedo is always darker. As analyzed in Sec. C.1, the joint inverse problem is ill-posed so there exists no unique solutions. Third, when the surface is primarily Lambertian, the rendered images can lose high frequency information. As a result, the recovered light map is also blurry. The converged roughness tends to be smaller than GT to compensate for this bluriness.

**SG Shading.** On the other hand, Spherical Gaussian shading has similar properties. We show optimization results in Fig. E and all the observations in MC shading can be applied to SG shading. More precisely, the optimized light always looks blurry while the optimized surface roughness is smaller compared to GT. This is also reflected in the learning task, as mentioned in Sec. 5.3 of the main paper.

# D  Dataset Overview

**Blender Car Dataset.** As mentioned in the main paper, we collected 485 different car models from *TurboSquid*[3] and 438 freely available high-dynamic range (HDR) environment maps from *HDRI Haven*[4]. For both two datasets we split the training set and testing set to around 4:1(400 cars for training, 85 cars for testing and 358 envmaps for training, 80 envmaps for testing). To render all 3D models, we use Blender's Cycles[5] path tracer with the Principled BRDF model [1].

We created two datasets on car models: Metallic-Surfaces and Glossy-Surfaces, as shown in Fig. K and Fig. L, respectively. For metallic surfaces, we set $\beta = 0$ and $m = 1$. Conversely, we set $m = 0$, $s = 1$ and randomly pick $\beta \in [0, 0.4]$ to generate images for glossy surfaces. For each dataset, the car is rendered in 24 fixed views, where we uniformly rotate the camera around. All the 3D models are normalized to be $[-18, 18]$ with camera distance and elevation to be 60 and $27°$. We randomly combine 3D models and light and render each car with 3 different environment maps.

---

[3]`https://turbosquid.com`
[4]`https://hdrihaven.com`
[5]`https://blender.org`

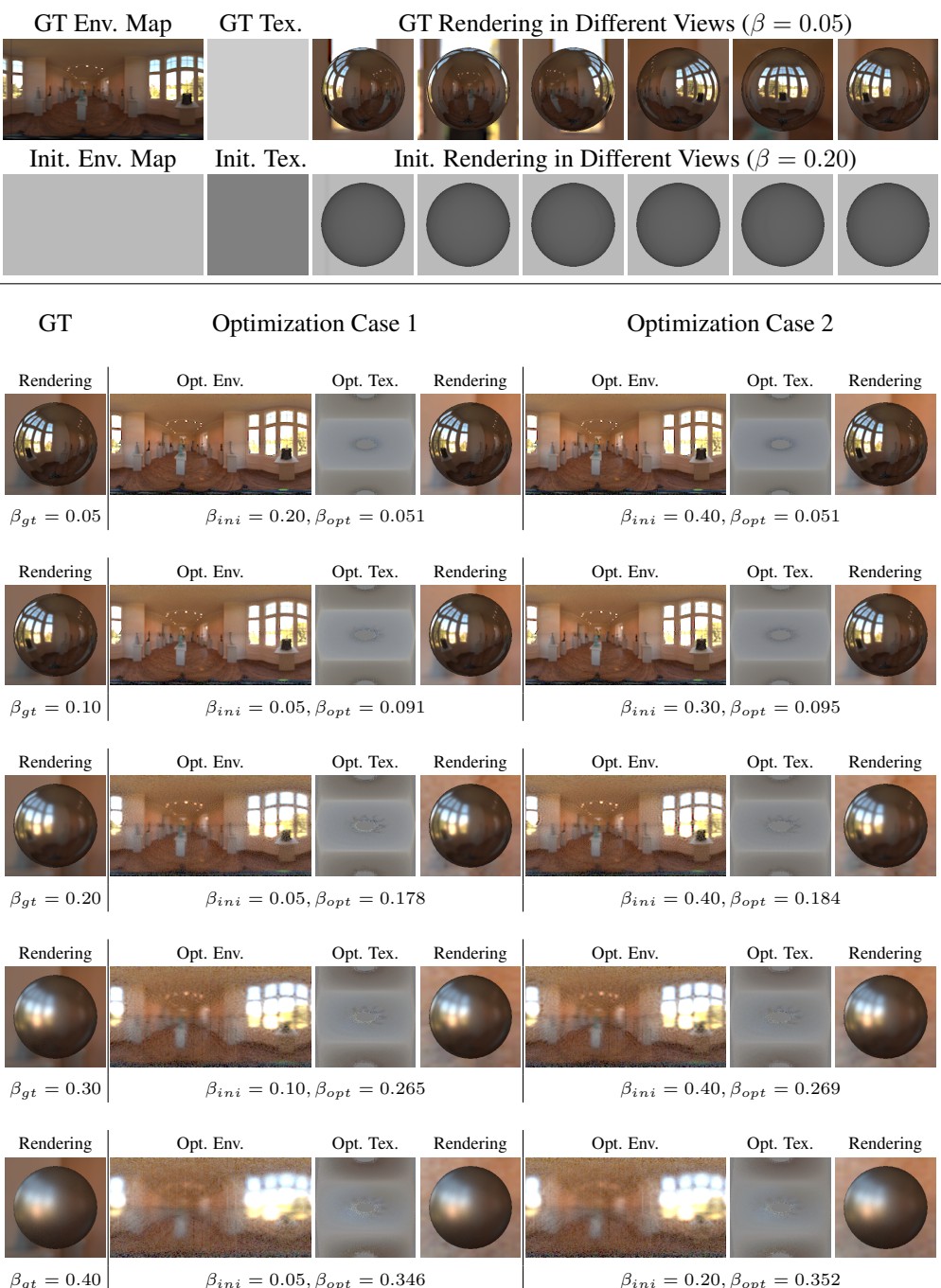

Figure D: **Monte Carlo Shading Optimization Results.** We fix the shape and optimize the environment map, diffuse albedo and surface roughness jointly from a foreground image loss only. To validate the convergence of different surface materials, we use the same environment map and albedo initialization but employ different roughness initialization and optimize for different GT roughness settings. **Top:** In the first two rows, we show the GT and initialized environment map, texture and rendering in 6 views. **Bottom:** In each row we optimize for one specific roughness, and in each column block we start with different roughness initialization. **Comments:** We successfully optimize all the parameters such that the rendered images (Col. 4 & 7) are very close to the GT rendering (Col. 1). However, due to the ill-posedness of the problem, the converged environment map, diffuse albedo roughness are not exactly the same as GT.

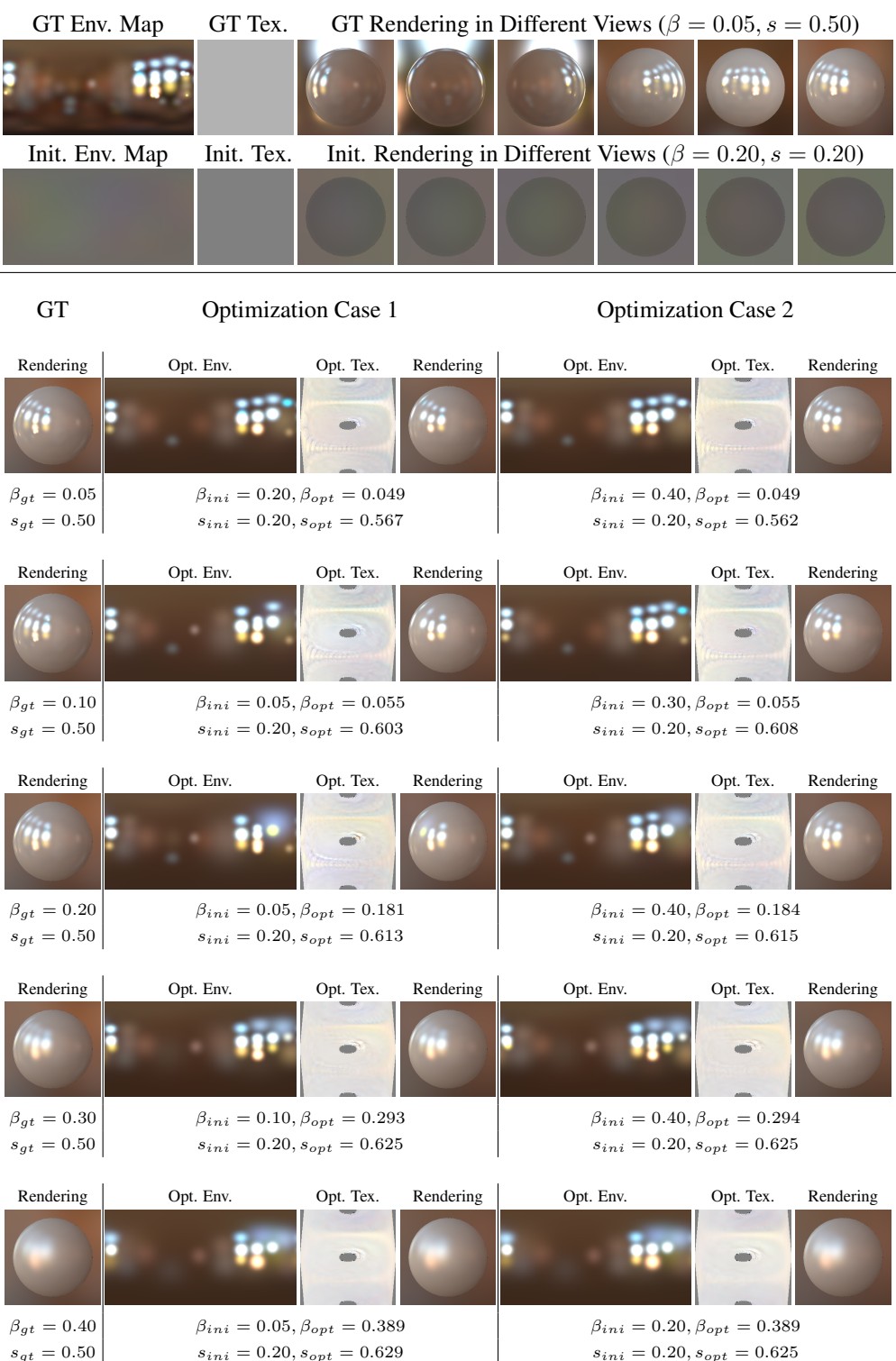

Figure E: **Spherical Gaussian Shading Optimization Results.** We fix the shape and optimize the environment map and surface material properties (roughness and specular albedo) jointly from a foreground image loss only. To validate the convergence of different surface material, we use the same environment map and texture initialization but different material initialization and optimize for different GT material settings. **Top:** In the first two rows we show GT and initialized environment map, diffuse albedo and rendering in 6 views. **Bottom:** In each row, we optimize for a specific GT material setting while in each column block we start from different surface material initialization. **Comments:** We successfully optimize all the parameters such that the rendered images (Col. 4 & 7) are very close to the GT rendering (Col. 1). However, due to the problem is ill-posed. the converged environment map, diffuse albedo and roughness are not exactly the same as GT.

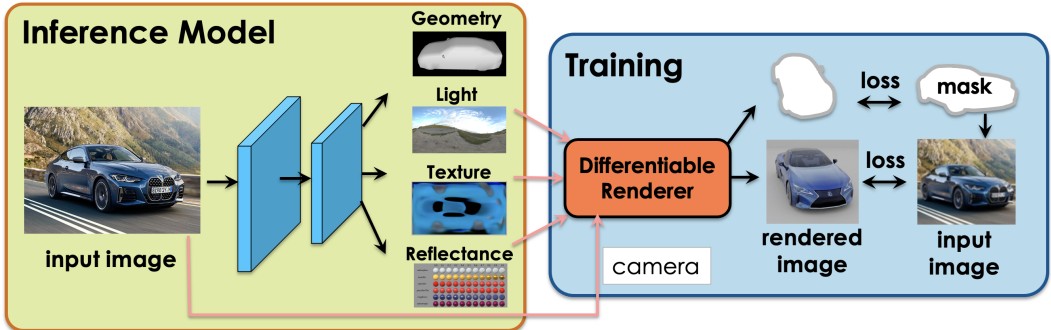

Figure F: **Training Pipeline.** given an image, we employ a neural network to predict all 3D attributes. We then adopt DIB-R++ to render them back to the images and supervise via the loss between the rendered image and input image. This model can be trained without any 3D supervision.

**Blender Face Dataset.** Besides two specular surfaces, we further use 515 (415 for training and 100 for testing) high-quality face scans from the *Triplegangers* dataset[6] (around half of them are white) to render roughly diffuse surfaces, as shown in Fig. M. We shade the face models with Cycles' random walk subsurface scattering for the skin, which is primarily diffuse in appearance. Since the choosen face models do not have hair, we render each model in 12 fixed front views. We use similar camera parameters and render each face with 3 random environment maps.

**GAN Dataset.** Following [14], we utilize StyleGAN to generate data to train a 3D reconstruction model. In particular, if we modify the first four latents (view point latent) and fix the remaining latents (content latent), StyleGAN can produce images of the same car from different view points. Conversely, fixing the view point latent and altering different content latent dimensions allows StyleGAN to produce images with different cars from a fixed angle. We use this strategy to generate 10 000 cars with 39 view point, and 10 000 faces with 14 different view points. Since these images contain complex surface roughness distribution, we apply SG shading to jointly predict shape, lighting and material from image supervision only. The overview of car dataset can be found in [14] while we provide face dataset overview in Fig. N.

# E    Additional Experimental Details and Results

We provide the details on the evaluation metrics in the main paper, and additional experimental results with failure cases on all the experiments in the main paper. We also include the experiments on diffuse surfaces, and additional comparisons with DIB-R [2] using SG shading.

## E.1    Single Image 3D Reconstruction Pipeline

We show a pipeline figure for the single image 3D reconstruction task in Fig. F, where we embed DIB-R++ into a learning framework and adopt neural networks to predict 3D properties without 3D training data.

Specifically, given an image, we first employ a neural network to predict all 3D attributes, including the geometry (sphere deformation), light (environment map), and texture (a spatial varying diffuse albedo map), and surface reflectance material (a global $\beta$ and $s$). We can then adopt a differentiable renderer to render them back to images and supervise via a loss between the rendered image and input image. This model can be trained without any 3D supervision, assuming good priors on geometry, light and materials.

## E.2    Multi-view Consistency

Since we have multi-view images for the same car, similar to [14, 2], we apply a multi-view consistency loss during training, which is the key to force the lighting to be separated from the texture. Specifically, we predict geometry/light/material in one view and supervise it in another view. When the input image has strong directional lighting effects, the optimization understands it should come from lighting instead of the texture, otherwise it would be wrong in the second view.

---

[6]https://triplegangers.com

Mathematically, let us assume we have two images $I_1$ and $I_2$ from two different poses (cameras noted as $V_1$ and $V_2$) of the same car. We can then predict shape, texture, surface reflectance and lighting for each image, noted as $S_1, T_1, M_1, L_1$ and $S_2, T_2, M_2\ L_2$. The multiview consistency loss is given by:

$$
\begin{aligned}
\mathcal{L}_{\text{im}} =& \|I_1 - \text{Render}(V_1, S_1, T_1, M_1, L_1)\|_1 + \\
& \|I_2 - \text{Render}(V_2, S_1, T_1, M_1, L_2)\|_1 + \\
& \|I_1 - \text{Render}(V_1, S_2, T_2, M_2, L_1)\|_1 + \\
& \|I_2 - \text{Render}(V_2, S_2, T_2, M_2, L_2)\|_1.
\end{aligned}
\tag{8}
$$

We predict shape, texture and material in canonical views while predicting light in image space. Besides the image loss, there are no more consistency regulations, i.e., we did not add any loss term to force $S_1$, $S_2$ or $T_1$,$T_2$ to be the same. Moreover, while more views would provide more constraints, we find that two views are enough. We thus use two views in all our experiments.

### E.3 Evaluation Metrics

Since GT lighting $\tilde{\gamma}$ is represented by an environment map while the predicted lighting may be in the form of spherical harmonics [2], spherical Gaussians or HDR texture envmaps with varying resolutions, we cannot directly compute $L^1$- or $L^2$-losses. Therefore, we evaluate them with normalized cross correlation (NCC). In particular, we convert the predicted lighting profile to an RGB image $\gamma_{\text{pred}}$ using an equirectangular projection and calculate the loss as:

$$
\mathcal{L}_{\text{NCC}} = 1 - \text{NCC}(\tilde{\gamma}, \gamma_{\text{pred}}), \qquad \text{where} \qquad \text{NCC}(\tilde{\gamma}, \gamma_{\text{pred}}) = \frac{\sum \tilde{\gamma} \odot \gamma_{\text{pred}}}{\|\tilde{\gamma}\|_2 \|\gamma_{\text{pred}}\|_2}
\tag{9}
$$

Similarly, for material predictions, different *uv*-parameterizations between GT and our predictions prevents direct comparison. Moreover, the predicted texture might be of different intensity scales compared to the GT. Thus, we first render the GT 3D models and predicted 3D models into images without lighting, and then apply NCC loss to the rendered albedo images as well.

In summary, the image loss is computed via the $L^1$ loss between the GT images and predicted images. 2D IoU loss is computed via the GT mask and predicted silhouettes [5]. We apply $\mathcal{L}_{\text{NCC}}$ to the GT lighting and predicted lighting as the light loss and apply $\mathcal{L}_{\text{NCC}}$ to the GT albedo and predicted albedo as the texture loss.

### E.4 Training Time

DIB-R++ contains a second shading stage. As a result, it is slower than [2] and [14] which only adopt rasterization. We find all the methods converge in 200 000 iterations, where [2, 14] cost 57 hours while DIB-R++ cost 77 hours. While longer, it is still affordable.

### E.5 Metalic and Glossy Surfaces

**Additional Results.** In the main paper, we validate our methods on metallic and glossy surfaces. We provide more results in Fig. O and Fig. P. Fig. O shows when the surfaces are highly reflective, MC shading successfully recover clean albedo maps and accurate light map, while [2] "bakes in" the environment map with texture. Similarly, Fig. P shows when the surfaces contain strong specular transport, SG shading separates them from the diffuse albedo while [2] merges them with texture (e.g., its textures contain incorrect brighter regions in the roof and front cover).

**Failure Cases**. We also provide failure cases in Fig. O and Fig. P. We find that when the GT environment map or the base texture map contains high frequency patterns, the problem becomes harder and our neural networks can only reconstruct low frequency content. This problem might be addressed with recent advances of position encoding and we leave it for future work.

### E.6 Shape and Material Evaluation

2D IoU loss measures the projection of the predicted shapes. As shown in Tbl. 1 in the main paper and Tbl. C, the scores of all the methods are pretty close, which indicates DIB-R and DIB-R++ have similar performance for the geometry. We also evaluate the Chamfer distance between the predictions

and GT meshes (normalized as one) in our glossy dataset predictions, where DIB-R++ is 0.036 while DIB-R is 0.037. It further demonstrates the both methods can recover good geometry.

We also check the predicted surface reflectance materials and GT and found little correlation. However, though the values are different, we still generate close shading effects to GT, as iun Fig. J. This indicates material and light disentanglement may have more than one solution, as evidenced in Fig. E and Fig. J.

### E.7 Additional Experiments on Diffuse Surfaces

We further apply all the methods to the purely dif-fuse/Lambertian surfaces and show the results in Fig. Q and Tbl. C. Unsurprisingly, both proposed models achieve unstructured lighting predictions. Since a diffuse BRDF has a large lobe and does not provide any view-dependent cues to improve the pre-diction of lighting, both MC and SG cannot accu-rately recover incoming radiance. On the other hand, since SH accounts for area light, it is most suitable for diffuse lighting prediction and it has slight better NCC score for lighting. As shown in Fig. Q, all meth-ods have smooth lighting predictions. As for texture

| Shading | Diffuse surfaces (↓) | | | |
|---|---|---|---|---|
| | Image | 2D IoU | Light | Tex. |
| MC | 0.030 | 0.032 | 0.109 | 0.039 |
| SG | 0.032 | 0.032 | 0.122 | 0.040 |
| SH [2] | 0.030 | 0.035 | 0.101 | 0.043 |

Table C: Quantitative results of single image 3D Reconstruction on diffuse surfaces. All methods have similar performance on texture prediction while SH has slightly better light prediction since it accounts area light.

predictions, most methods seem to have different intensities to GT (SG and MC are lighter while SH is darker) to accommodate to the lighting and materials. However, the re-rendered images are close to GT.

**Failure Cases**. We also provide a failure case in the bottom of Fig. Q. Unsurprisingly, we find that when the dataset is biased, the trained model is also biased. *Triplegangers* dataset is mostly composed of Caucasian human faces and thus the model can fail for darker skin tones. Thus, our method is not suitable real-world deployment. Additional situated analysis should be carefully performed to determine if DIB-R++ is acceptable to use in domains that potentially impact human lives.

### E.8 Real-world Dataset

We provide more qualitative results in Fig. R and Fig. S for the prediction on Cars and Faces images generated from StyleGAN [14], respectively. Our DIB-R++ faithfully reconstructs better material and lighting components compared to [14]. In particular, our model can represent the dominant light direction more accurately, while naive shading [14] tends to merge reflectance with lighting.

Lastly, we also apply our model to real images in LSUN dataset [13] in Fig. T. Our model is trained on synthetic dataset [14] generated by StyleGAN [4]. Thanks to the powerful generative model, the distribution of GAN images is similar to the distribution of real images. Thus, our model can be well generalized to real images. We show additional reconstruction results of LSUN [13] real images in Fig. T.

## F Discussion and Ablation Study

We provide ablation study to evaluate the performance of our DIB-R++. We first analyze the choice of rasterization renderer in Sec. F.1, then compare SH [2] and SG shading in Sec. F.2, then explore the effects of numbers of SG components in Sec. F.3. We then ablate the loss terms and analyze the specular effects in Sec. F.4 and Sec. F.5. Finally, we run our model three times and report the training variance to show that our method is insensitive to random seed.

### F.1 Choice of Differentiable Rasterizer

Our DIB-R++ rendering framework is quite flexible. In the first stage, it supports an arbitrary differentiabler rasterization method, e.g. NVDiffRast [6] or DIB-R [2]. Compared to DIB-R, NVDiffRast is more efficient. However, the way NVDiffRast handles occlusion relies on anti-aliasing, while DIB-R uses soft rasterization. Theoretically, anti-aliasing only influences edge pixels while soft rasterization computes the soft probability of all pixels, providing more gradient signals in

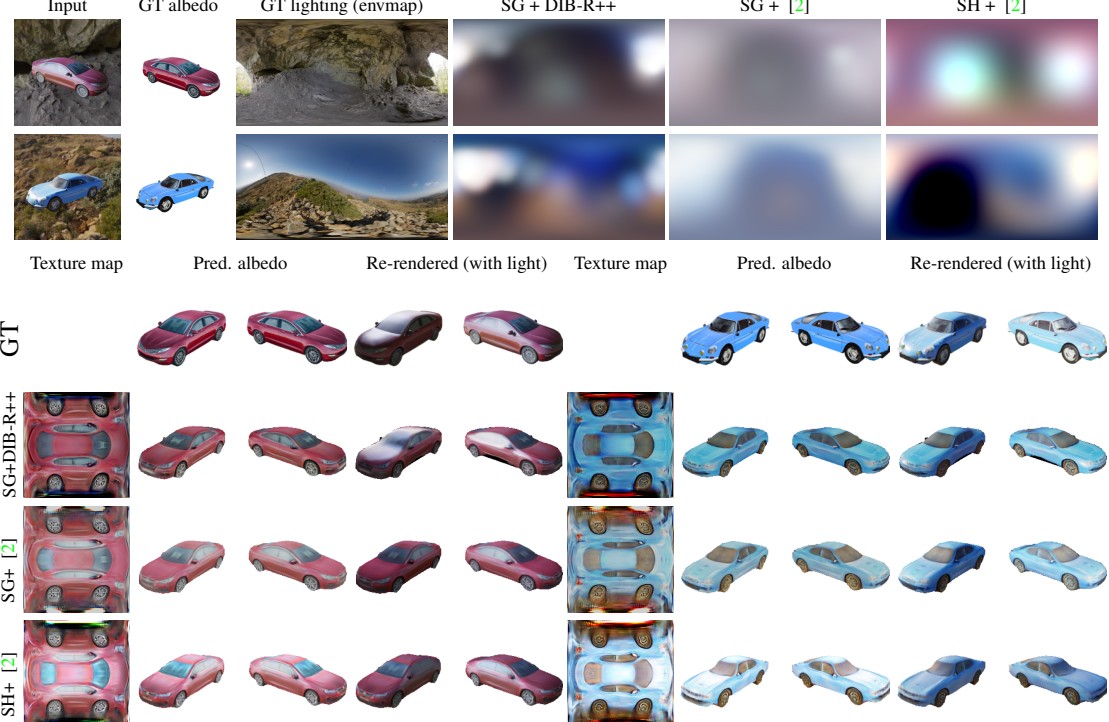

Figure G: **Ablation Study of Shading Methods.** Even though we apply the same lighting representation to [2], due to the lack of specular transport suport, [2] cannot perfectly disentangle the specular light from diffuse labedo. Its window and front cover still "bake in" white specular light. Moreover, the lighting is also blurry, as opposed to DIB-R++. This demonstrates the necessity of explicitly modelling advanced lighting effect in the renderer.

shape deformation. We experimentally found that the predicted shape from DIB-R is better than NVDiffRast so we chose DIB-R in the paper.

## F.2    Comparison of SH [2] and SG Shading

DIB-R++ achieves much better disentanglement for reflective surfaces compared to [2], since our model can account for surface reflections while [2] only supports naive diffuse surfaces low-frequency lighting. We have shown quantitatively and qualitatively comparisons with [2] in the main paper, where [2] uses Spherical Harmonic (SH) lighting while we use Spherical Gaussian lighting. However, among these results, we use the default lighting for [2], which uses a 2-order SH basis and contains 27 parameters ($9 \times 3 = 27$, 9 for 2nd SH and 3 for RGB). On the other hand, DIB-R++ applies SG shading and 32 components SG light, which is composed of 224 parameters ($32 \times 7 = 224$). As such, DIB-R++ differs from [2] in two key aspects: shading method and light representations. Thus, we further apply the same SG lighting to [2] and compare with DIB-R++ again, where the only difference is the shading method to perform a more fair comparison.

We show the comparison results in Fig. G and Tbl. D. After applying SG lighting to [2], it is still incapable of disentangling strong lighting from the base texture. As a result, the car's window and front cover still "bake in" white specular light. Moreover, the lighting prediction of [2] is blurrier. Quantitatively, the lighting loss of [2] with SG is better than that of [2] with SH, probably due to more parameters (224 vs. 27). However, it is still worse than

| Shading | Glossy surfaces ($\downarrow$) | | | |
|---|---|---|---|---|
| | Image | 2D IoU | Light | Tex. |
| DIB-R++ w/ SG | 0.024 | 0.057 | **0.091** | **0.140** |
| [2] w/ SG | 0.023 | 0.063 | 0.098 | 0.147 |
| [2] w/ SH | 0.024 | 0.062 | 0.131 | 0.152 |

Table D: Comparison of shading methods.

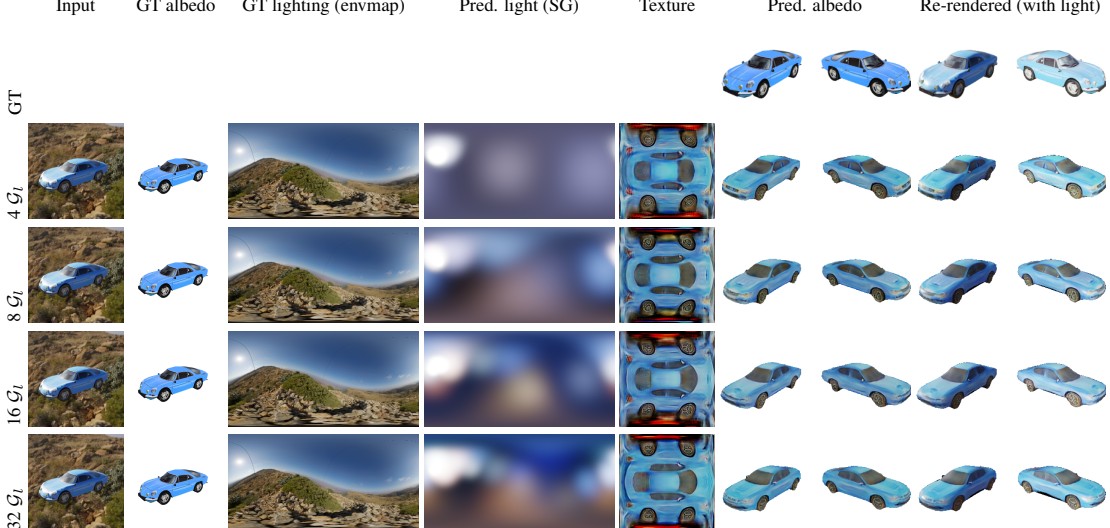

Figure H: **Ablation Study of Number of SG Components.** As the number of SG components increase, the predicted lighting is more and more similar to the GT lighting and the color of predicted texture becomes more and more visually uniform, which indicates better disentanglement.

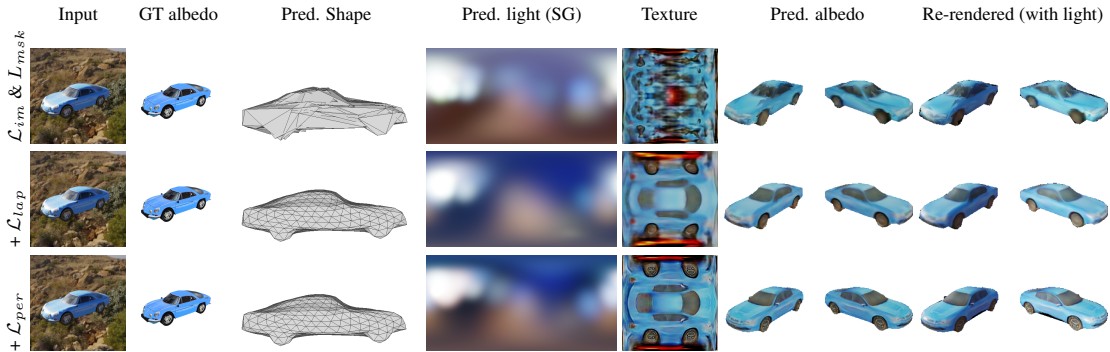

Figure I: **Ablation Study of Loss Terms.** Similar to [14], we also find $\mathcal{L}_{\text{lap}}$ regularizes the geometry to be smooth while $\mathcal{L}_{\text{per}}$ helps create high-frequency, detailed textures.

DIB-R++ with SG. This demonstrates the necessity of explicitly modeling advanced lighting effects in the renderer.

### F.3 Number of SG Components

We study the influence of the number of SG components $K$ and show results in Fig. H and Tbl. E. When SG components are less than 4, the representation ability is limited. Consequently, the predicted lighting only has one mode. It is also reflected in Tbl. E: less components generally result in higher lighting prediction error.

On the other side, when the SG components are larger than 16, the performance becomes similar (Tbl. E, 16 vs. 32). However, more SG components have higher representation ability and predicted visually better results, as shown in Fig. H. Thus, we generally choose $K = 32$ in the experiments to make full use of a 16GB memory GPU.

| Shading | Glossy surfaces ($\downarrow$) | | | |
|---|---|---|---|---|
| $K$ SGs | Image | 2D IoU | Light | Tex. |
| 4 | 0.025 | 0.067 | 0.118 | 0.147 |
| 8 | 0.025 | 0.063 | 0.093 | 0.145 |
| 16 | 0.024 | 0.063 | 0.092 | 0.143 |
| 32 | 0.024 | 0.057 | 0.091 | 0.140 |

Table E: Ablation study of number of SG components.

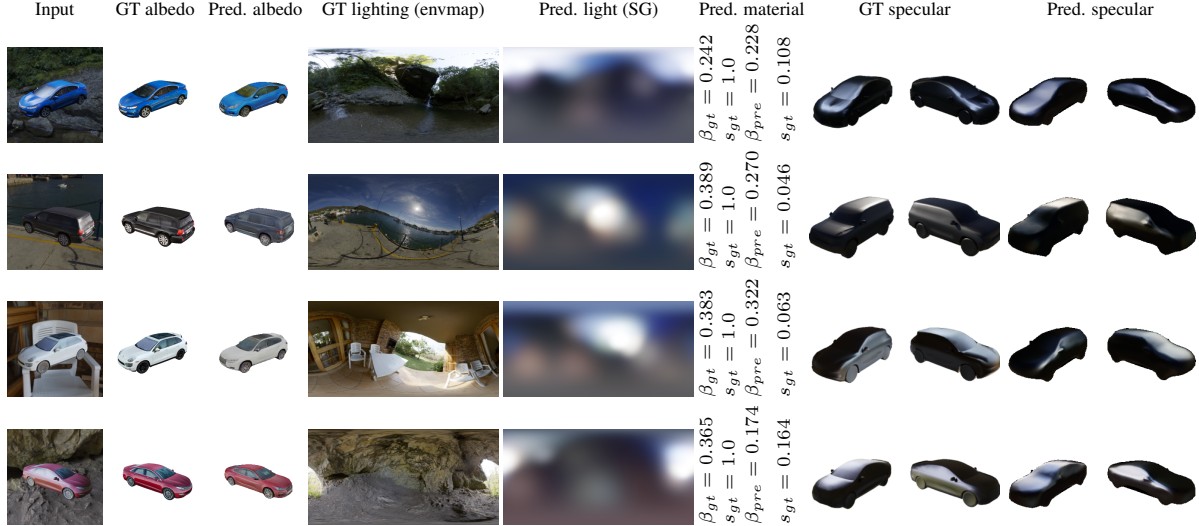

| Input | GT albedo | Pred. albedo | GT lighting (envmap) | Pred. light (SG) | Pred. material | GT specular | Pred. specular |

Figure J: **Specular effect of SG Shading.** We decompose the prediction into diffuse albedo, light and material parameters, then visualize the specular contribution in other views and compare with GT (last 4 columns). Although the predicted material parameters do not match GT, interestingly, the specular renderings align well with each other.

## F.4 Ablation Study of Loss Terms

We also ablate the loss terms in our method in Fig. I and get similar observations as in [14]. Specifically, we find $\mathcal{L}_{lap}$ regularizes the geometry to be smooth while $\mathcal{L}_{per}$ helps in creating high-frequency, detailed textures.

## F.5 Specular Effects of SG Shading

We also decompose the prediction into the albedo map, lighting and material parameters, and display the specular effect in other views in Fig. J. Compared to GT, we find the specular effects align very well with each other, even though the predicted material parameters has little correlation with GT. It might indicates the material and light disentanglement may have more than one solution, as also evidenced in Fig. E.

## F.6 Training Variance

Lastly, we train our model three times on glossy surfaces and report the performance in Tbl. F. Our model converges well, and random seeds had little effect on performance. The models have very similar predicted results and close scores. The variance is small, which demonstrates the robustness of our model.

| Shading | Glossy surfaces ($\downarrow$) | | | |
|---|---|---|---|---|
| | Image | 2D IoU | Light | Tex. |
| DIB-R++ w. SG round 1 | 0.024 | 0.057 | 0.091 | 0.140 |
| DIB-R++ w. SG round 2 | 0.025 | 0.061 | 0.092 | 0.143 |
| DIB-R++ w. SG round 3 | 0.025 | 0.062 | 0.091 | 0.143 |
| Mean | 0.025 | 0.060 | 0.091 | 0.142 |
| Variance | 0.000 | 0.002 | 0.000 | 0.001 |

Table F: Training variance.

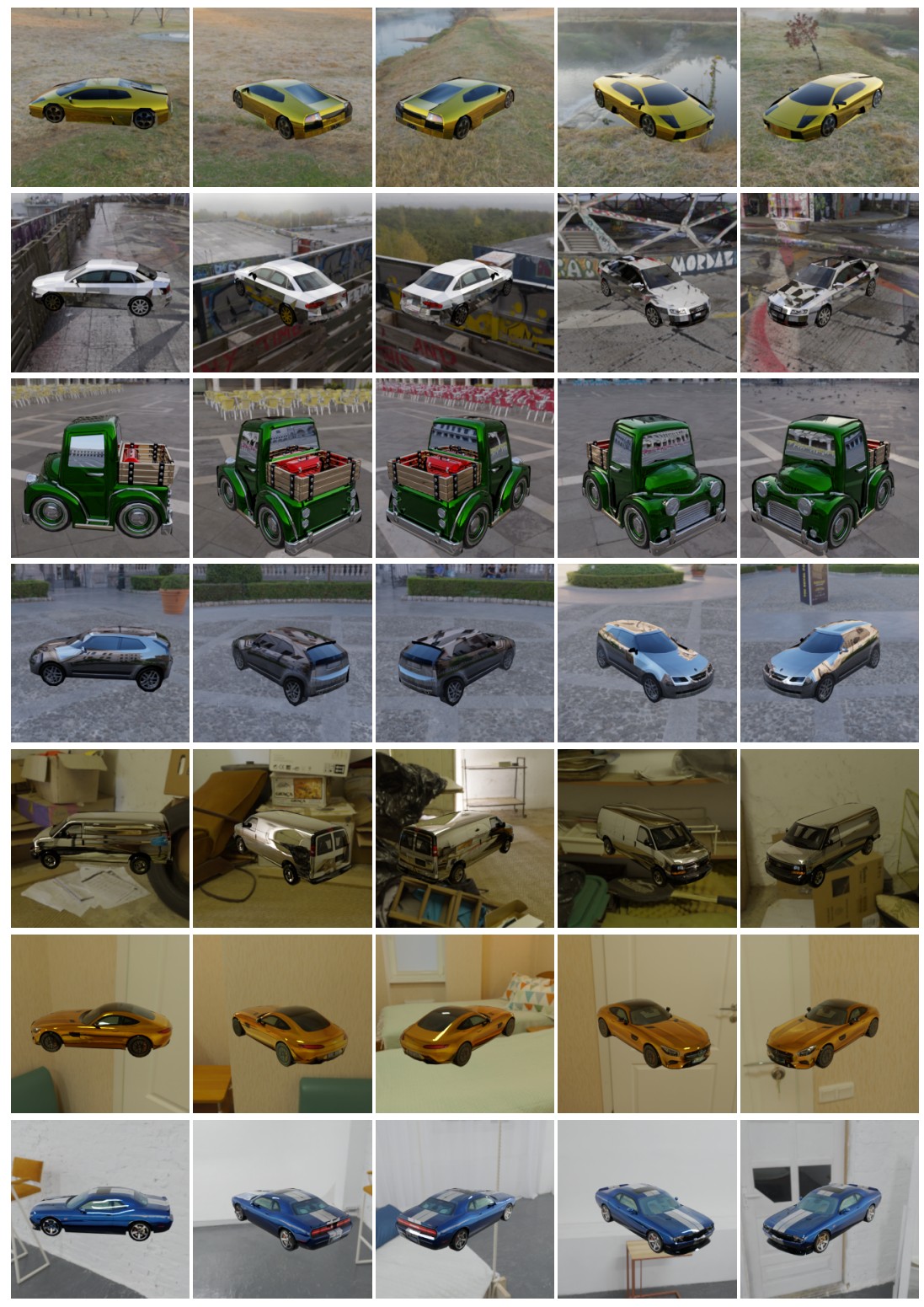

Figure K: **Metallic Surface Dataset Overview**. We render various cars under both indoor and outdoor environment maps. We set the $m$ as 1 and $\beta$ as 0 and the rendered images are highly reflective. The reflected environment maps can be observed in the car body, which provide strong signals in lighting estimation.

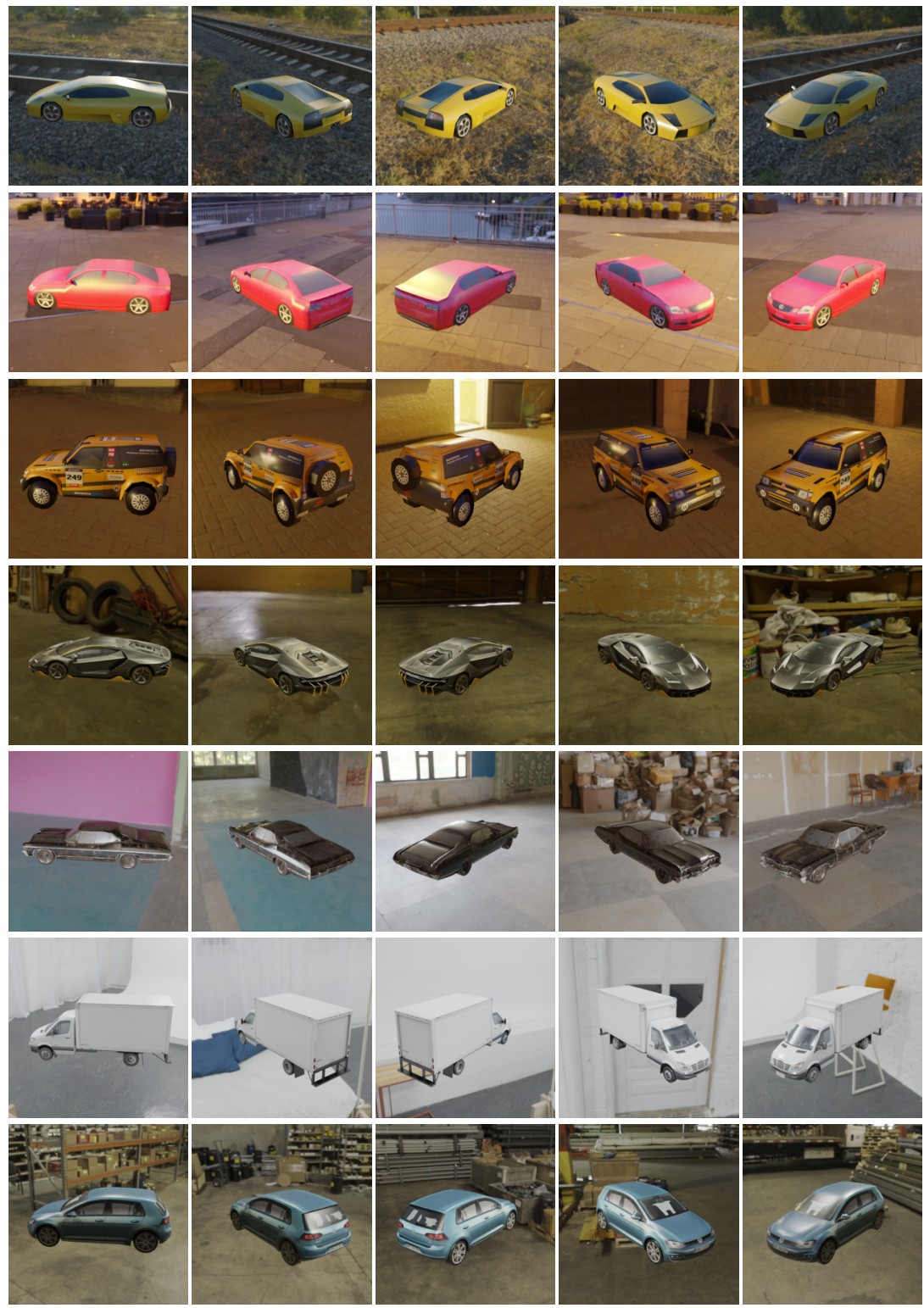

Figure L: **Glossy Surface Dataset Overview.** We render various cars under both indoor and outdoor environment maps. We set the $m = 0$ and vary $\beta \in [0, 0.5]$ so that the rendered images contain view-dependent highlights.

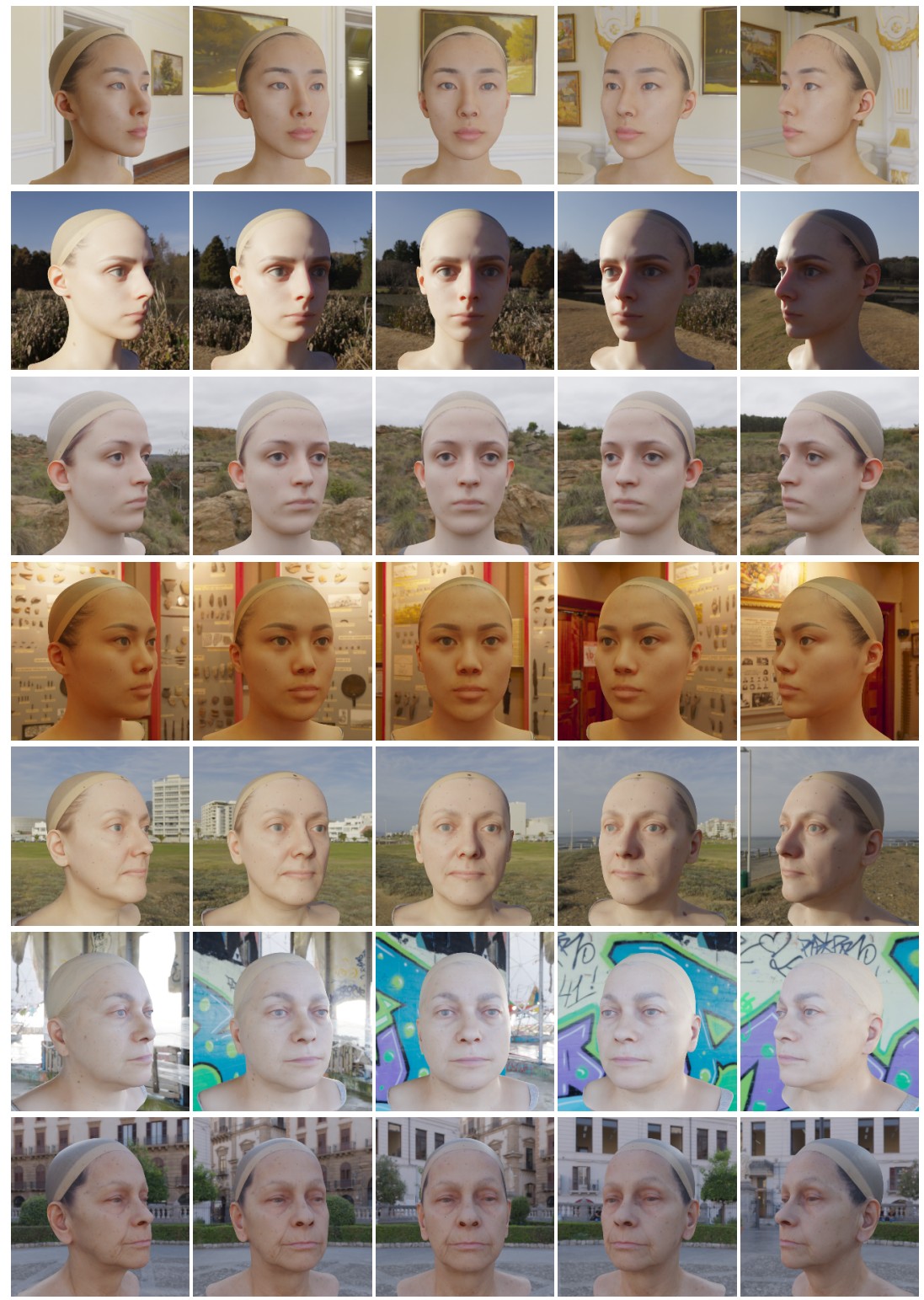

Figure M: **Diffuse Surface Dataset Overview.** We render various 3D female face models. We use Blender Cycles' random walk subsurface scattering for the skin. The rendered images look mostly diffuse.

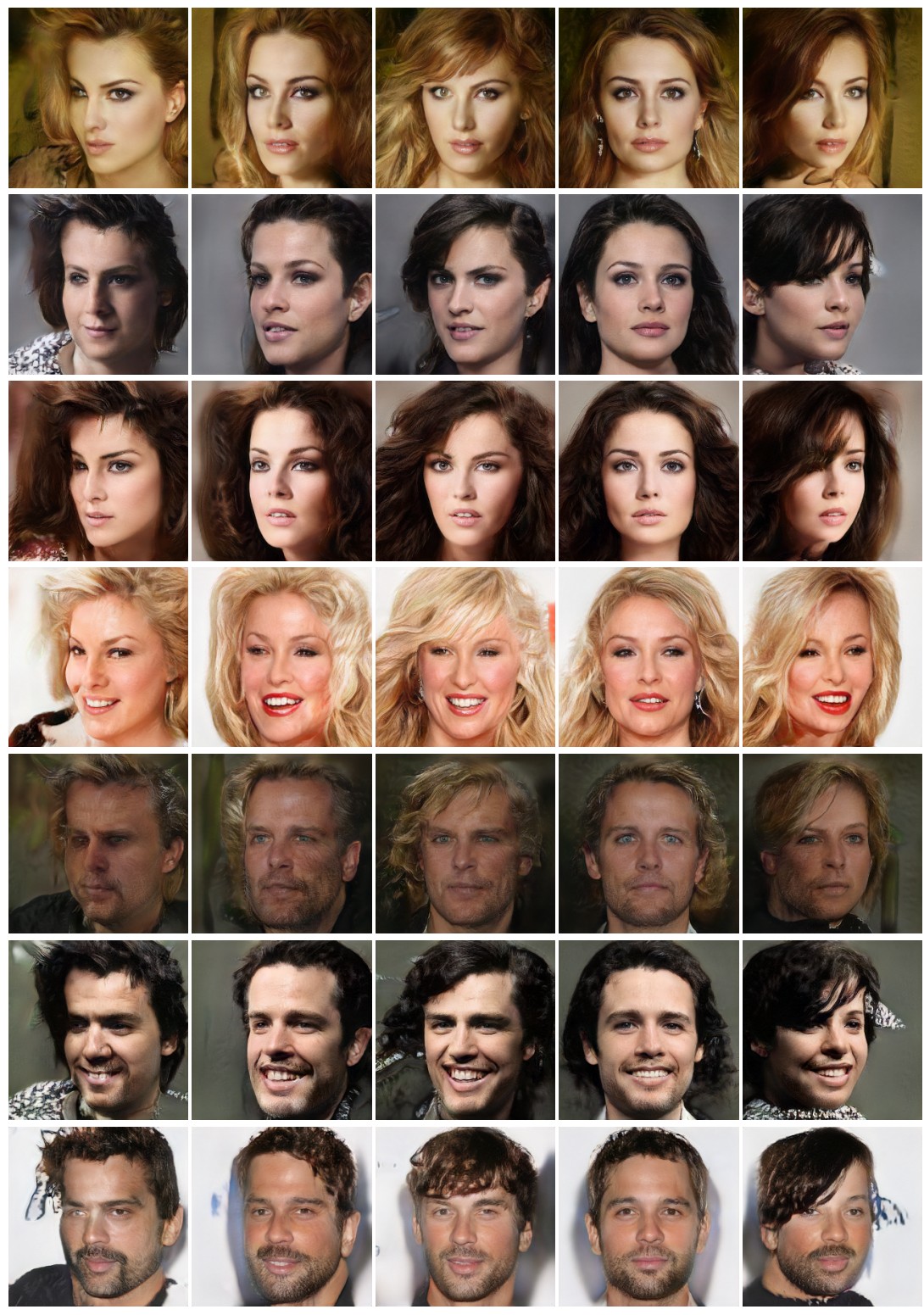

Figure N: **Realistic Face Dataset Overview.** We synthesize multi-view datasets for faces following the pipeline introduced in [14]. The generated dataset contain objects with various shapes, textures and viewpoints. Following the observation in [14], we generate consistent poses of object in each column.

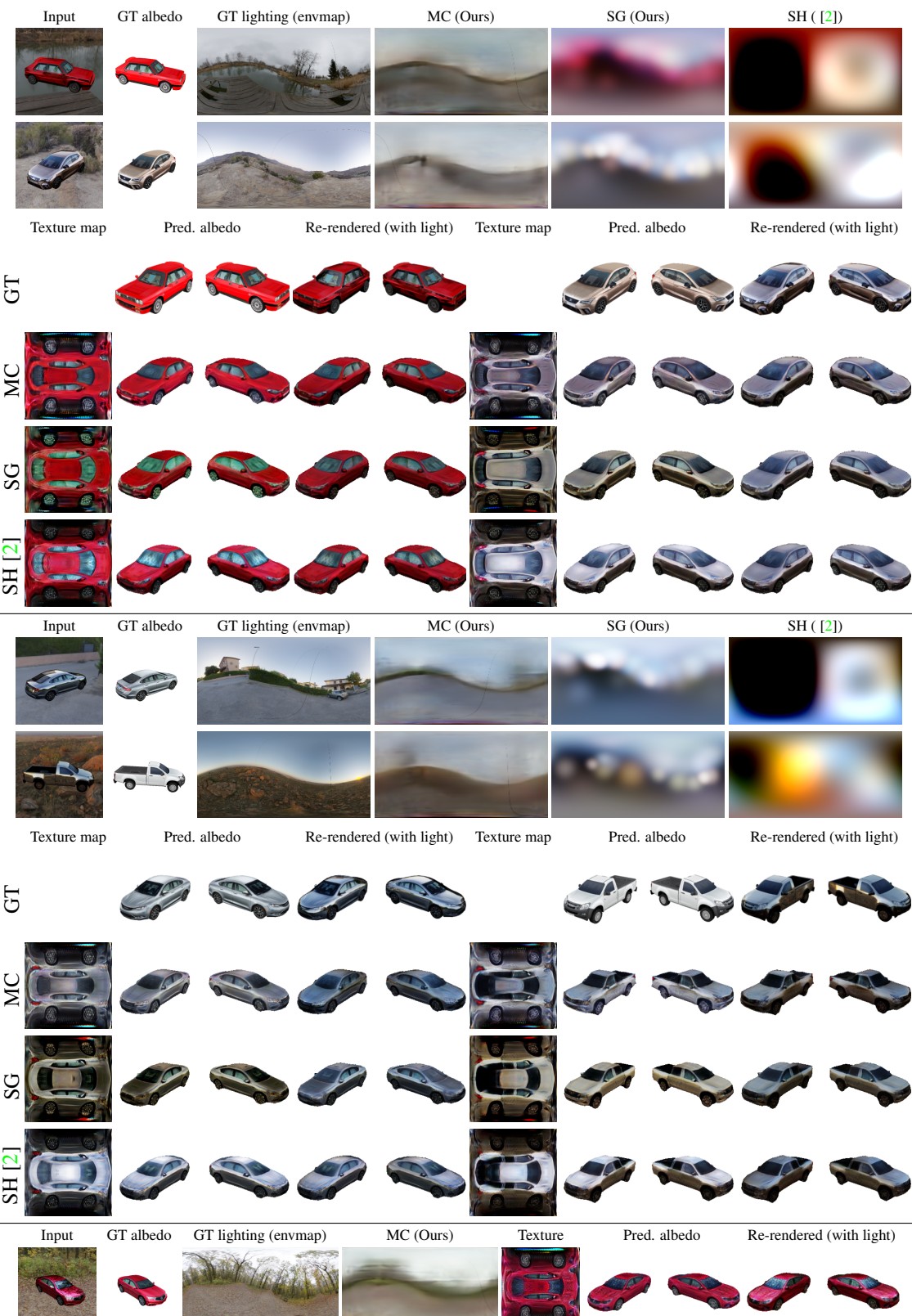

Figure O: **Prediction on the Metallic Car Dataset** ($\beta = 0$). **Top and Middle:** We show more comparison results for 3 different shading methods. While all re-rendered images look almost identical, the underlying radiometric components (material and lighting) are different. A SH basis [2] cannot recover the high frequency details of the sky light maps. In this case, MC performs best due to low variance in the estimator for mirror-like BRDFs. SGs can recover the overall form and contrast of the light map, but tend to predict incorrect texture maps incorporating the ground dominant color. **Bottom:** We show a failure case in the bottom row. MC shading fails when the environment map contains a lot of high frequency details. It reconstructs low frequency content but also merges the high frequency leaves into the texture.

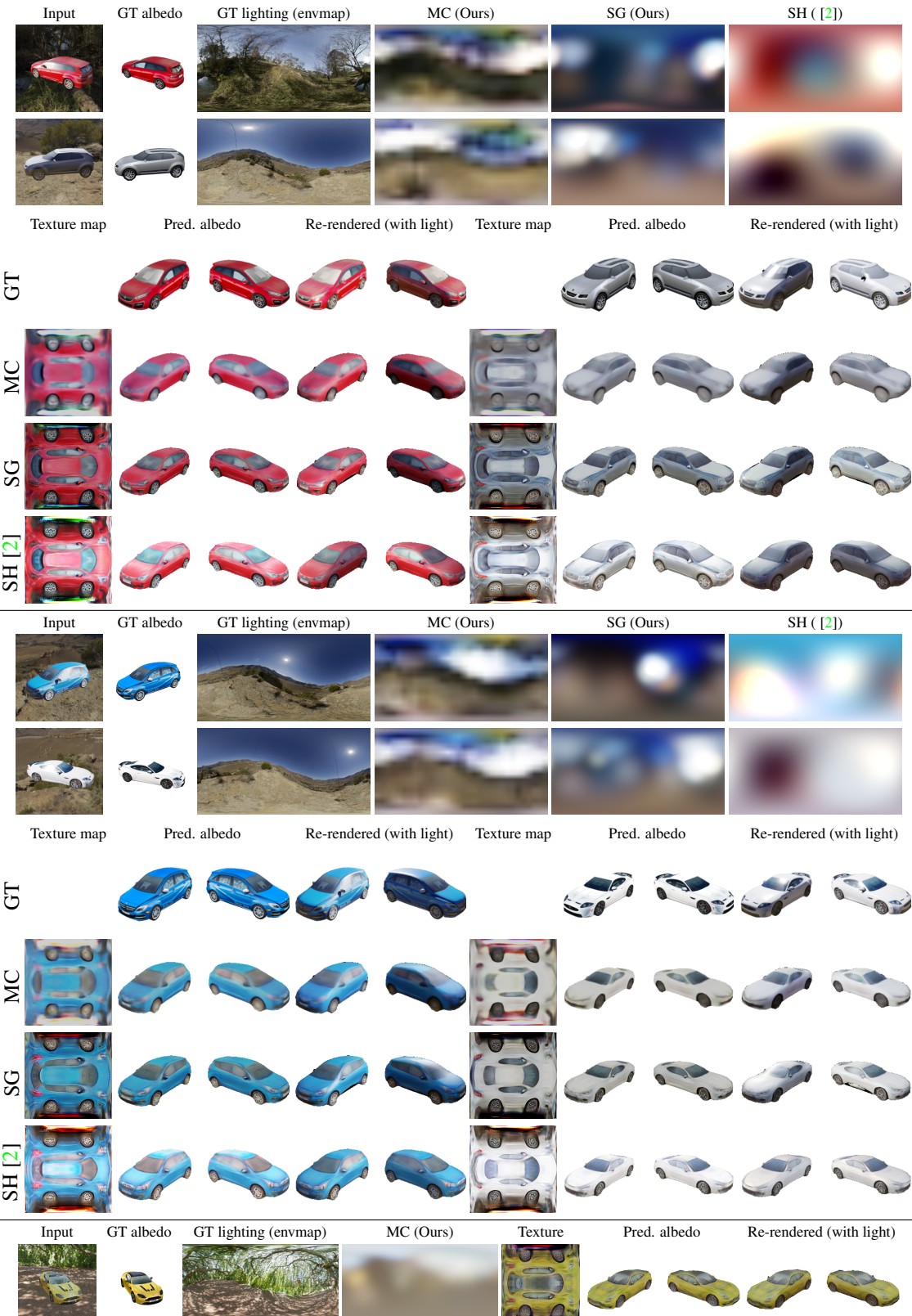

Figure P: **Prediction on the Glossy Car Dataset** ($\beta > 0$). **Top and Middle:** We show more comparisons for three different shading methods. When the objects are glossy but not perfectly specular, SGs can correctly disentangle reflectance from lighting, as evidenced by the absence of white highlights in the predict albedos. Chen et al. [2] cannot capture these bright regions due to a diffuse-only shading model, while MC oversmooths the predictions due to noisier pixel gradients. While all methods cannot completely reconstruct the environment map, our method can predict the correct dominant light location and sky color. **Bottom Row**: We show a failure case in the bottom row. All the methods fail when the texture contains a lot of high frequency details. We show SG only reconstructs low frequency content.

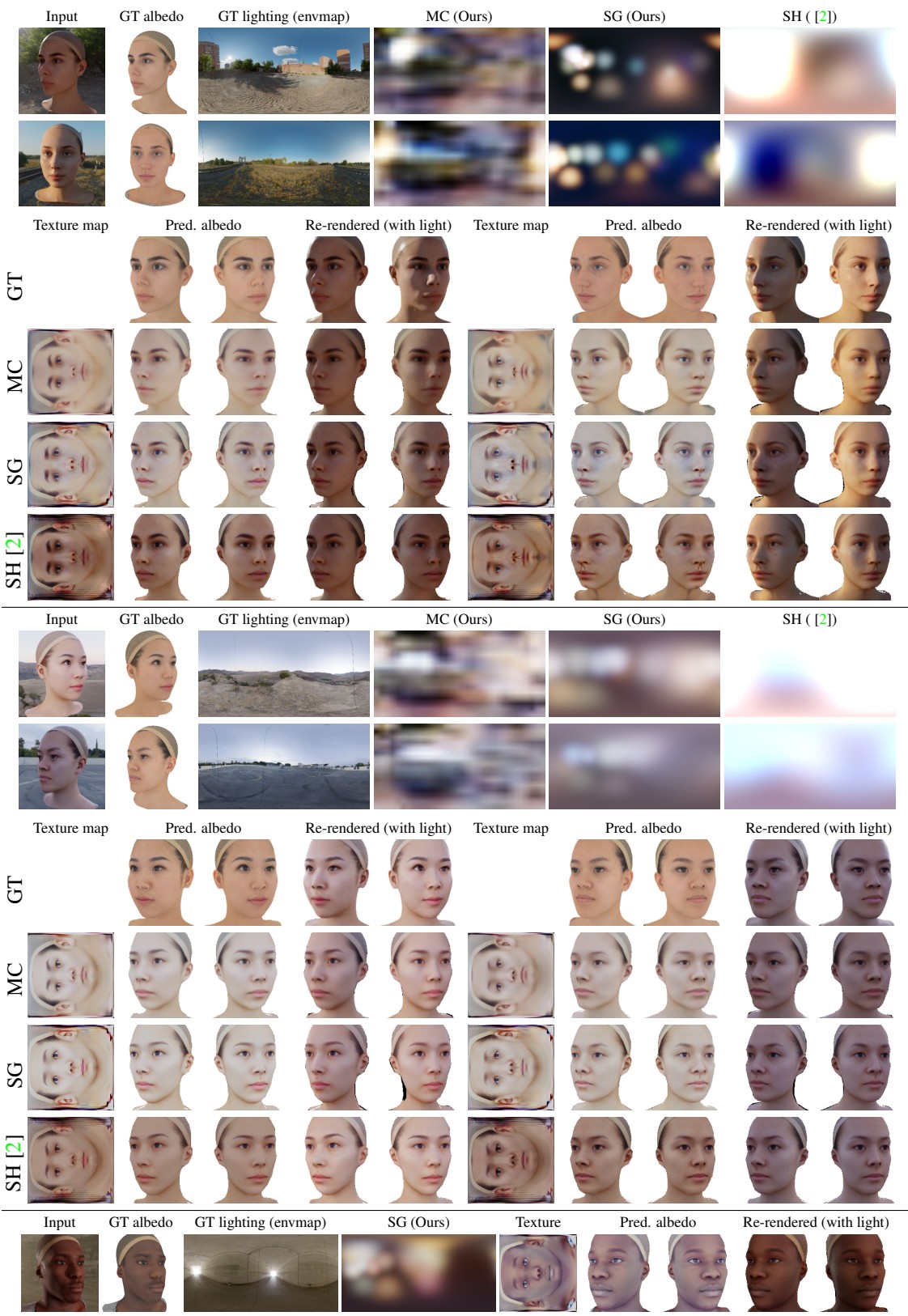

Figure Q: **Prediction on the Diffuse Face Dataset** ($\beta \approx 0$). **Top and Middle:** We show comparisons for three different shading methods. When the objects are mostly diffuse, all methods have unstructured smooth lighting prediction and similar texture maps. **Bottom Row:** We show a failure case for black skin. Due to the inherent white face bias in our dataset, the predicted albedo regrettably tends to be too light, although the re-rendered images closely match reference.

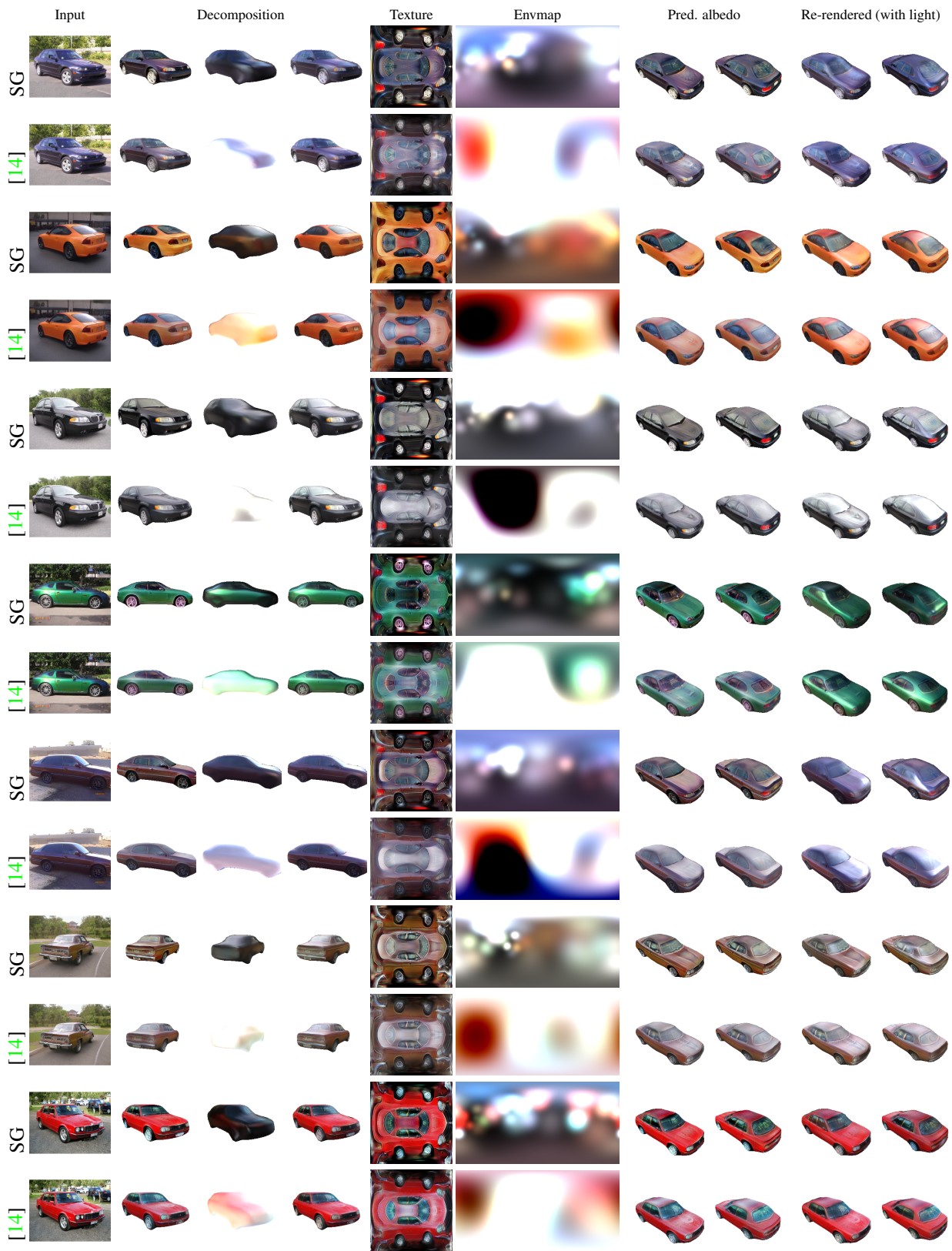

Figure R: **Prediction on Real-world Dataset (Cars).** DIB-R++ accounts for high specular light and always have cleaner textures compared to [14].

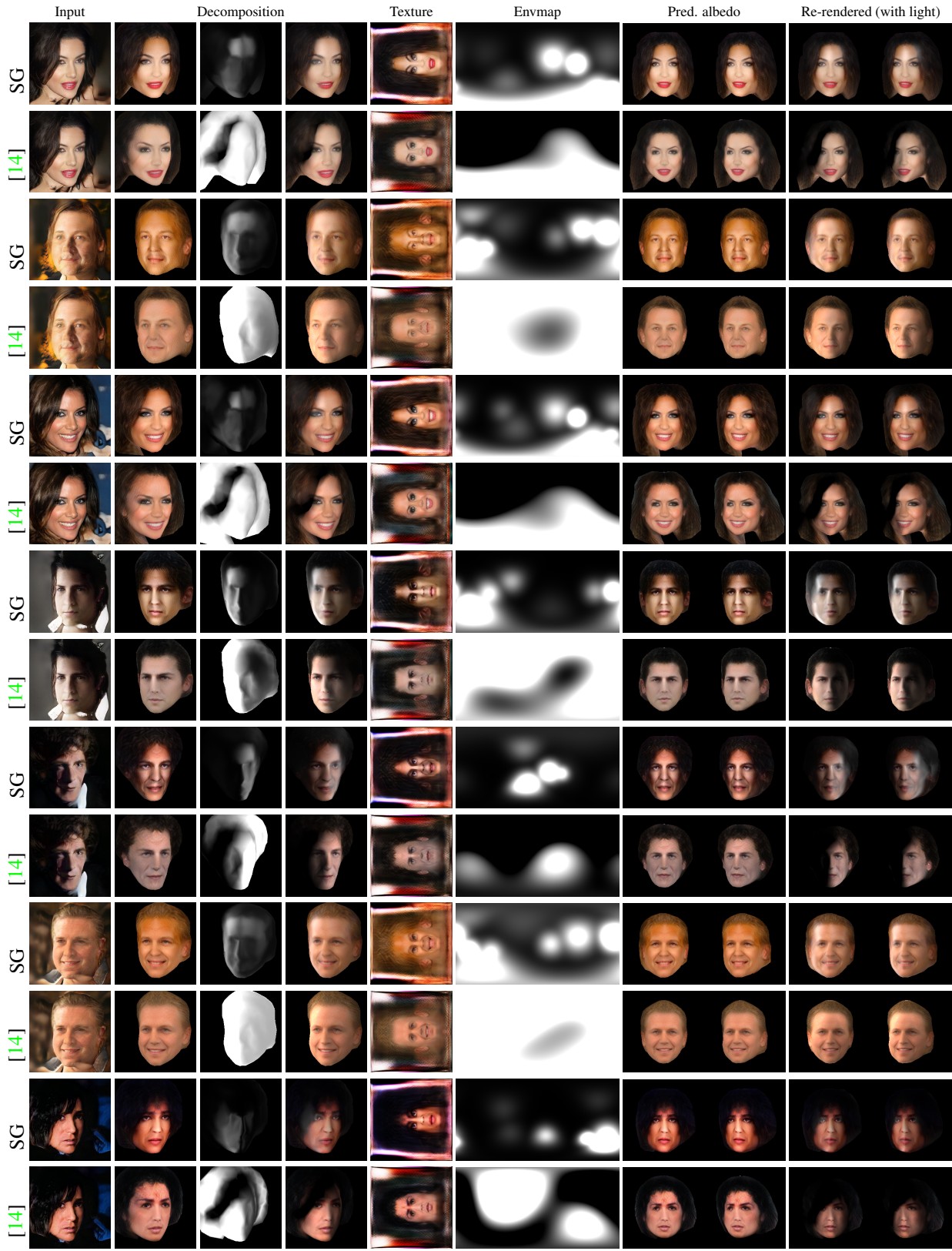

Figure S: **Prediction on Real-world Dataset (Faces)**. DIB-R++ accounts for high specular light and always have cleaner texture compared to [14].

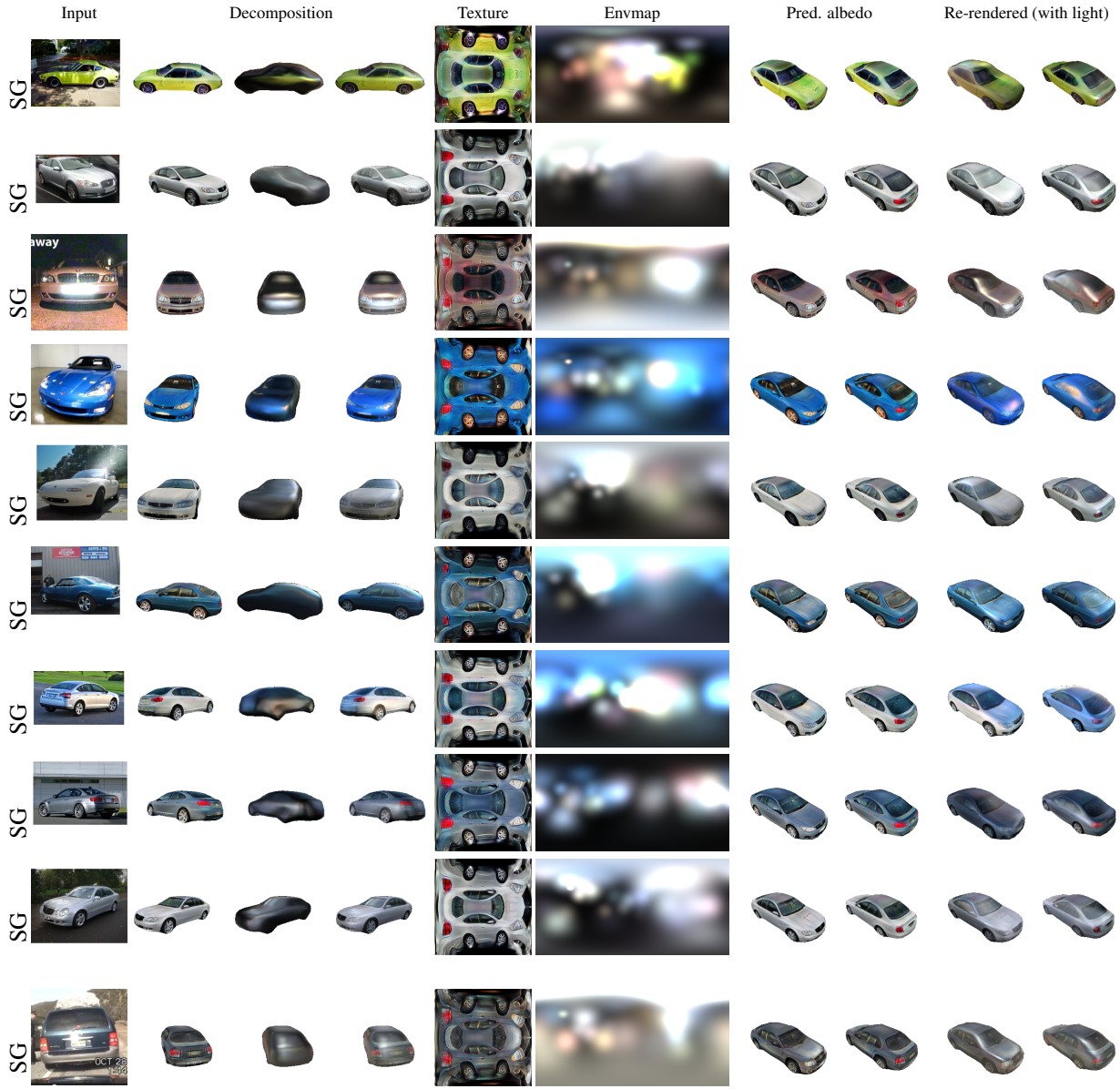

Figure T: **Prediction on LSUN Dataset (Cars).** DIB-R++, trained on StyleGAN dataset, can be well generalized to real images. Moreover, it also accounts for high specular light and predict correct lighting directions and clean textures.