# OpenReview forum: "DIB-R++: Learning to Predict Lighting and Material with a Hybrid Differentiable Renderer"
_NeurIPS.cc/2021/Conference — NeurIPS 2021 Poster_

### Official Review · Reviewer_ShTU · 2021-07-12

**Rating:** 5
**Confidence:** 4

**Summary:**

This paper proposes an improved pipeline for self-supervised training of a neural network that recovers the 3D properties (geometry, brdf, light) of an image containing a single object. This is achieved by proposing DIB-R++, a differentiable renderer which offers improved rendering speed at the cost of less accurate renderings. The DIB-R++ builds on top of the differentiable rasterization framework of DIB-R in order to add physically-based material and lighting models. DIB-R++ comes in 2 variations: A) MC -  relies on Monte Carlo sampling of incident light directions, which is better suited for reflective surfaces. B) SG - relies on using Spherical Gaussian for fast analytic integrations, which is better for rough surfaces. It is important to note that DIB-R++ assumes direct illumination only.

Experimentations include analyzing the effects of forwards and backward passes through the DIB-R++ renderer in isolation (figures 2 and 3), and training neural networks on metal synthetic (section 5.1), glossy synthetic (section 5.2), and StyleGAN generated (section 6.1) datasets. This paper also contains interesting experiments to demonstrate the effects of using different variations and parameters of this renderer in terms of speed and rendering quality in the supplementary materials.

**Ethical Concerns:**

One of the evaluations of this paper is on inferring the 3D properties (geometry and reflective parameters) of human faces. The authors have done a good job of acknowledging the biases that may be imposed on their models from the training data (section 7, broader impact). Since the face images from the dataset shown in the original paper/supplementary materials do not represent diversity of ethnicities very well, the decomposition might not produce correct results for other ethnicities (I have witnessed such failure cases in the past). However, it should be noted that recovering particularly human face 3D parameters is not the primary concern of this paper.

**Ethics Review Area:**

["Discrimination / Bias / Fairness Concerns"]

**Limitations And Societal Impact:**

I would prefer if the fact that the training phase relies on multi-view images was stated more clearly, especially in the conclusion/limitations. This was briefly mentioned in the experimental settings of section 6.1. Also might want to mention that this approach is only suitable when we want to recover the geometry and material of a single 3D object captured in an image.

**Main Review:**

Pros:
+ This paper is well presented and experiments were nicely detailed.
+ The experiment comparing using SG vs MC for the DIB-R++ rendering (figure 2) offers interesting insights on the suitability of each renderer based on the material roughness.
+ The optimization task (figure 3) is reassuring that differentiating through DIB-R++ yields meaningful signals with respect to the material and lighting conditions
+ Models trained with DIB-R++ seem to recover better light and material parameters compared to previous works.

Cons:
- Contribution of paper should be stated more clearly - initially the renderer formulation (DIB-R++) seems to be the contribution, but I can not pinpoint the novelty from a graphics/rendering point of view as all of the used formulations were pre-existing and cited. After further reading, the main contribution seems to be in evaluating how the formulation of DIB-R++ fits into the context of training neural networks.
- There is a claim of disentangling lighting and materials in the title and abstract of this paper, but I'm not sure what in the proposed formulation of DIB-R++ would cause for better disentanglement? Seems to be a better renderer for training neural networks in terms of less computations for better performance, but there isn't really anything in the formulation to encourage disentanglement. This is also hinted at by the authors in supplementary materials section C.1. Therefore I find the disentanglement term misleading.

Questions:
- How does DIB-R++ compare to other differentiable renderers, e.g. the one from Mitsuba 2?
- How does the training time of the models using DIB-R++ renderer compare to the methods of "Learning to predict 3D objects with an interpolation-based differentiable renderer" by Wenzheng Chen et al. and "Image GANs meet differentiable rendering for inverse graphics and interpretable 3D neural rendering" by Yuxuan Zhang et al.? Is it slower or faster to train? Does the model converge in less or more iterations?

Overall, I like the experimentations of this paper and they demonstrate improved performance compared to previous methods. However, the contribution in the renderer does not seem significant enough to me for a NeurIPS paper.

**Needs Ethics Review:**

Yes

**Time Spent Reviewing:**

10

---

> ### Author Response · Authors · 2021-08-10
> **Answers to Reviewer ShTU**
>
> We thank the reviewer for detailed comments. We address all the concerns below.
>
> >Contribution of paper should be stated more clearly
>
> Thanks for the suggestion. Please see the novelty part in the shared comments. We believe our contribution lies in two parts. 1) introducing SG and MC in the learning framework and 2) being the first method to predict geometry, advanced lighting and SVBRDF from a single image in an unsupervised way. We will clarify our contributions in the revision.
>
> >...what in the proposed formulation of DIB-R++ would cause for better disentanglement
>
> Previous differentiable rendering methods assume naive lighting and Lambertian surface. As a result, when the input image contains advanced lighting effects, the predicted textures of  [7] or [11] often have specular light ‘’bake in’’ the textures.
> DIBR++ solves this problem by introducing advanced BRDF and physically based rendering equations which account for such lighting effects.
>
> Another reason is we have multiview images for the same object and adopt multiview consistency loss during training, which force the lighting to be separated from the texture. For example, we predict geometry/lighting/material in one view and supervise it in another view. When the input image has specular effects, the rendering formula knows it should contribute to lighting instead of texture. if not, it would be wrong in another view.
>
> As shown in Fig. 5 & 6, our predictions are much cleaner and better disentangled from lighting. That’s why we claim our framework has a better disentanglement. we will add the discussion in a revision.
>
>
>
> >How does DIB-R++ compare to other differentiable renderers, e.g. the one from Mitsuba 2?
>
> While Mitsuba 2 provides more complex multiple bounce rendering, the complex designs prevent it from being deeply integrated in the learning frameworks(e.g. training neural networks with mitsuba2). Instead, our method, written by PyTorch, can be easily integrated within neural networks.
>
> Furthermore, since both our method and Mitusba2[3] supports ray tracing, we also evaluate the running time and memory for the same ray tracing optimization task. We restrict the ray samples to be the same (4 samples) and run 300 times iterations to optimize the environment map while fixing everything else.  Our method performs faster (5.4sec v.s. 9.8sec) and requires less memory(1475MB v.s. 3287MB), which demonstrates the advantage of our renderer.
>
> >How does the training time of the models using DIB-R++ renderer..
>
> DIBR++ contains a second shading stage, which is slower than [7] and [11] using DIBR. We find all the methods converge in 200,000 iterations, where [7,11] cost 57 hours while DIBR++ cost 77 hours. While it is longer, it is still affordable.
>
> >...the training phase relies on multi-view images was stated more clearly, especially in the conclusion/limitations….this approach is only suitable when we want to recover the geometry and material of a single 3D object captured in an image.
>
> Thanks for the suggestion. We will make it more clear about the multi-view training data requirement and the single object reconstruction task in the limation.

---

### Official Review · Reviewer_FLZ6 · 2021-07-16

**Rating:** 6
**Confidence:** 3

**Summary:**

DIB-R++ proposes an extending DIB-R with a physics based shading model. The method predicts an environment map for lighting and uses a simplified version of isotropic Disney BRDF model. The shading model is single bounce and is estimated using either Monte Carlo methods or using spherical basis functions. With these modifications, the authors demonstrate that they are able to decompose the lighting and materials of single objects (cars, faces) from a single image better than DIB-R.

**Limitations And Societal Impact:**

The authors do a good job of explaining the limitations throughout the paper.

**Main Review:**

The paper is well written and well explained. The addition of more physically based shading to DIB-R seems like a logical and well motivated extension. The paper presents an interesting analysis of the pros and cons of using Monte Carlo versus Spherical Gaussians. From a computational complexity standpoint, the spherical gaussians approach makes a lot of sense and the qualitative results confirm that the impact on quality is minimal. I have listed some suggestions/questions/concerns below, overall I think this paper is marginally above the acceptance threshold.

a) It is noted in LN 251 that DIB-R can accurately re-render the image (albeit with incorrect albedo and lighting). From the quantitative and qualitative results it is fairly easy to conclude that DIB-R++ produces more physically plausible results, however, the usefulness of this improvement is never demonstrated to the reader. It would be informative to see the modifications in Fig 7 applied to DIB-R to hopefully demonstrate why this more complex method should be considered.

b) Topics like glossy/metallic and MC/SG are well analyzed in the paper, but other topics like geometry are barely discussed. It would be nice to know how the proposed modifications to DIB-R affect the predicted geometry. Additionally, it would be good to include metrics computed against the ground truth geometry. Table 1 is also missing a number of details, specifically what are these values actually measuring (Mean squared error?), please specify. Also consider reporting other common image metrics such as PSNR, SSIM, and LPIPS.

c) The paper relies heavily on the qualitative results. The way they are currently presented makes it difficult to determine the effectiveness of the method. It would be beneficial to include examples of the ground truth albedo in the same poses as the predicted albedo example. Additionally it would be good to include ground truth examples of the scene re-rendered (with light).

Other:
+ Why is approximating the cosine foreshortening with a SG useful?
+ LN 291. Can you provide these errors, I think it would be interesting to the reader to understand how far off these are.

**Time Spent Reviewing:**

4

---

> ### Author Response · Authors · 2021-08-10
> **Answers to Reviewer FLZ6**
>
> We thank the reviewer for detailed comments. We address all the concerns below.
>
> >It would be informative to see the modifications in Fig 7 applied to DIB-R to hopefully demonstrate why this more complex method should be considered.
>
> Thanks for the suggestion. In Fig.7 we demonstrate we can edit albedo and material with DIBR++. While material is not supported by DIBR, we believe DIBR++ produces cleaner texture which is more suitable for albedo editing, and DIBR bakes lighting into texture, making the rerendered images less realistic. We will add the examples of DIBR in the revision.
>
> >...it would be good to include metrics computed against the ground truth geometry. Table 1 is also missing a number of details, specifically what are these values actually measuring (Mean squared error?), please specify. Also consider reporting other common image metrics such as PSNR, SSIM, and LPIPS
>
> Thanks for the suggestion. We provide the 2D IOU loss in Tbl 1 in the paper and Tbl B in supp, which measures the projection of predicted shape. The scores are close to each other, which indicates DIBR and DIBR++ have similar performance of geometry. We also evaluate the chamfer distance between the predictions and GT meshes (normalized as 1), where DIBR++ is 0.036 while DIBR is 0.037. It further demonstrates both methods get pretty good shape recovery. We will add the chamfer comparison in the revision.
> In Tbl1, we measure MAE(mean absolute error) loss for image, 2D IOU loss for mask, normalized cross correlation loss(Eq 8 in supp) for albedo and lighting. We will clarify it and add more metrics in the revision.
>
> >It would be beneficial to include examples of the ground truth albedo in the same poses as the predicted albedo example. Additionally it would be good to include ground truth examples of the scene re-rendered (with light).
>
> Thanks for the suggestion. We will add GT abledo and GT scenes rendered with light for corresponding views in the revision.
>
> >Why is approximating the cosine foreshortening with a SG useful
>
> By approximating the cosine term with a SG, we can easily compute an analytic solution for the rendering integral (Eq 1). Without this approximation, one has to manually derive the integral of the lighting and BRDF SGs multiplied by a cosine term, which may not admit a closed form expression. Note that this approximation is less accurate at grazing angles, but we did not observe any significant degradation in quality.
>
> >LN 291. Can you provide these errors,
>
> We will add the results of the difference between the predicted materials and GT material in the revision.

---

### Official Review · Reviewer_8HSK · 2021-07-17

**Rating:** 5
**Confidence:** 5

**Summary:**

The paper proposes a method to reconstruct geometry, material and lighting from
a single image. They apply differentiable rasterizers to calculate the rendering loss,
which is back-propagated to supervise the training of the desired components.
They compare different lighting representations include HDR envmaps as well as
Spherical Gaussians.


**Limitations And Societal Impact:**

See main reviews.

**Main Review:**


1. My major concern over the paper is the technical contribution of the paper. The
paper builds on an existing rasterization-based inverse-rendering framework DIB-R.
The difference is that DIB-R assumes a Lambertian model, while the proposed method
handles specular materials. However, the overall framework is almost the same.
Jointly optimizing lighting, materials, and geometry are also common, for example,
[14].  The paper says [14] does not generalize to real images, and however [14] does
show results on real images and the paper does not compare to [14]. It is also not clear to
me why the proposed method would generalize, since it is also trained on synthetic dataset.
For me, it seems the major contribution of the paper is that it adds the lighting, material in
the inverse-rendering framework of DIB-R, which I think is incremental.

2. The paper has a lot of text on the shading model and lighting representation. While
I appreciate the efforts, I have to say that both representations are pretty standard in
computer graphics and such discussions do not increase the technical contributions.

3. In terms of differentiable rasterizers, I think [8] has better performance
than [7] in terms of better handling occlusions and efficiency, why not use [8]?

4. The loss function only consists of losses on images and vertices. Since the model
is trained on a synthetic dataset, why not also render ground truth material and lighting
and use them as direct supervisions? Involving those GT in training would better resolve
the ambiguity in inverse rendering.

5. The inverse-rendering problem itself is highly ambiguous. In the case that the object
is purely lambertian, there is no way to infer the lighting accurately, and such a problem
exists for both SGs and envmaps. Therefore the proposed method is mostly limited to glossy
surfaces.

6. In the results section, the paper provides no comparison against previous single-image lighting
and SVBRDF estimation method, for example [14]. For lighting estimation, the following paper
should be cited and compared: Deep Parametric Indoor Lighting Estimation, ICCV 2019.

7. In all results on synthetic data, the corresponding ground truth relighting images should
be included as references. Also there should be videos showing renderings under different lighting
conditions and viewpoints with comparison to ground truth.


Overall I think the technical contribution of the proposed method is not enough to be accepted.
Most of the components are from previous works and the rasterizer-based optimization framework itself is also standard. Therefore, I vote for rejection.



**Time Spent Reviewing:**

2

---

> ### Author Response · Authors · 2021-08-10
> **Answers to Reviewer 8HSK**
>
> We thank the reviewer for the feedback. We address all the concerns below. We believe there is a major misunderstanding that the reviewer has regarding our paper, and we wish to further clarify.
>
> The problem we are tackling is jointly predicting material, geometry, and lighting from a single image without any supervision, instead of doing it in a supervised manner, as acquiring ground truth data(for real images) is extremely hard. We achieve this goal by integrating DIBR++ in neural networks and training in an analysis-by-synthesis fashion. It is totally different from [14] and [ref1], which belong to supervised methods, as pointed in the shared comments.
>
> >My major concern over the paper is the technical contribution of the paper
>
> Please see the novelty part in the shared comments.
>
> >It is also not clear to me why the proposed method would generalize, since it is also trained on synthetic dataset.
>
> We apologize for the confusion and want to clarify that our method is trained without any 3D supervision and it can be applied to not only synthetic images (Fig.4 & Fig. 5), but also GAN-generated realistic images (Fig.6).  Following [11], we apply our method to realistic images generated by StyleGAN. [11] has shown the model trained on StyleGAN dataset can be well generalized to real images. This is also observed in our experiments, where we try car images in LSUN dataset and get plausible reconstructions. We will add such generalized real images reconstruction in the revision.
>
> >In terms of differentiable rasterizers... why not use [8]
>
> Our image based rendering framework is quite flexible. In the first stage, it supports any rasterization method, e.g. Nvdiffrast[8] or DIBR[7]. We agree that [8] is more efficient. However, the way Nvdiffrast[8] addresses occlusion relies on antialiasing, while [7] uses softrasizer. Theoretically, antialiasing only influences edge pixels while [7] computes the soft probability of all the pixels, providing more signals in shape deformation. We experimentally find the predicted shape form [7] is better than [8] so we choose [7] in the paper.  We will add the discussion in a revision.
>
> >...why not also render ground truth material and lighting and use them as direct supervisions?
>
> We agree that adding lighting and material supervision in the training would definitely improve the performance in the synthetic dataset. However, these direct supervisions come from either synthetic data with a domain gap or real-world data that are expensive to capture.
> Our paper mainly focuses on utilizing differentiable rendering to infer everything in an unsupervised way. Adding synthetic supervision makes it limited to synthetic images and becomes harder to generalize to real imagery.  We do observe a strong domain gap between synthetic images and real images, where the model trained on synthetic data always fails for real images.
>
> >Therefore the proposed method is mostly limited to glossy surfaces.
>
> We agree SG and MC have more advantages in specular cases. However, our method is not limited to glossy surfaces. We have shown the results of diffuse objects in Tbl.B and Fig N in supp, where MC, SG and SH have similar performances. It means we can still apply SG to diffuse objects.
>
> >...the paper provides no comparison against previous single-image lighting and SVBRDF estimation method, for example [14]. For lighting estimation, the following paper should be cited and compared: Deep Parametric Indoor Lighting Estimation, ICCV 2019.
>
> Although our paper, [14] and the mentioned ICCV paper are all doing single-image lighting and SVBRDF estimation, we want to clarify that [14] and ICCV paper requires lighting supervision while our paper, to the best of our knowledge, is the first method which recovers advanced lighting and materials without any lighting or materials supervision. As an unsupervised method, our approach is quite different from [14] and ICCV paper, and it is unnecessary to compare them two since the supervision is totally different.
>
> >...the corresponding ground truth relighting images should be included as references. Also, there should be videos showing rendering...
>
> Thanks for the suggestion. We will add GT albedo in all views and create corresponding videos in the revision.

---

> > ### Comment · Reviewer_8HSK · 2021-09-02
> > **Reviewer feedback**
> >
> > I thank the authors for the clarifications. I can understand the reason for not using synthetic ground truth. However, as with other reviewers, I am still concerned about novelty. I feel that the paper is putting different components together where each component has been explored previously. Therefore, I changed my rating to 5 and refrain from giving a higher score.

---

### Official Review · Reviewer_SWGM · 2021-07-28

**Rating:** 5
**Confidence:** 4

**Summary:**

This paper proposes a single image inverse rendering method, named DIB-R++ that extends DIB-R [7] with MC/SG-based ray-tracing to accelerate training/inference. This method works in two-stage, first it uses DIB-R [7] to infer the mesh (and associated geometric attributes) from a single input image, then in the second stage, the network infers albedo and environment light jointly, and a direct light only shader renders a photorealistic image from the inferred geometry, material and lighting.

**Limitations And Societal Impact:**

See Weaknesses above.

**Main Review:**


Strengths:

1. The proposed DIB-R++ is unsupervised and does not require ground truth of geometry, material or environment light.

2. In experiments, the proposed DIB-R++ outperforms DIB-R [7] on both metallic and glossy surfaces.


Weaknesses:

1. My major concern about this paper is novelty. Using SG to accelerate training and inference is not a new idea, and it has been studied in NeRD [33] and PhySG [34], the difference is that the proposed DIB-R++ uses DIB-R [7] to recover a mesh from the input image, then infers material and illumination (represented by MC or SG, which is not novel). So I think the novelty is rather incremental compared with both NeRD [33] and PhySG [34] and DIB-R [7].

2. The paper mentioned single image 3D reconstruction in Section 4, but there were no such experimental evaluations in the paper. It is hard to tell the 3D reconstruction quality without quantitative evaluations on reconstructed mesh, depth or normal.

3. This paper also lacks experimental comparisons with a state-of-the-art method, e.g., [ref1] Li et al. Inverse rendering for complex indoor scenes: Shape, spatially-varying lighting and svbrdf from a single image. CVPR 2020. Moreover, the proposed DIB-R++ can only infer direct illumination, while [ref1] can also learn global illumination. I'd like to hear more discussion on this in rebuttal.

4. The paper is not easy to follow, and it would be good to show a system diagram that describes network architecture, network input/output, data flow and training/inference steps.


**Time Spent Reviewing:**

4

---

> ### Author Response · Authors · 2021-08-10
> **Answers to Reviewer SWGM**
>
> We thank the reviewer for the detailed comments. We address all the concerns below.
>
> ## Clarification of summary
> We apologize for the confusion and want to clarify that our single image inverse rendering method is trained end-to-end and it jointly predicts geometry, albedo, lighting, and material in a single stage. Instead, the rendering pipeline contains 2 stages, where we first adopt rasterization to render mesh and texture into albedo and normal maps, then shade with different approaches(MC or SG). But the rendering pipeline is integrated within the end-to-end network framework.
>
> ## Novelty.
>
> Please see the shared comments.
>
> >It is hard to tell the 3D reconstruction quality without quantitative evaluations on reconstructed mesh, depth or normal
>
> Thanks for the suggestion. We provide the 2D IOU loss in Tbl 1 in the paper and Tbl B in supp, which measures the projection of the predicted shapes. The scores are close to each other, which indicates DIBR and DIBR++ have similar performance of geometry.
> We also evaluate the chamfer distance between the predictions and GT meshes (normalized as 1), where DIBR++ is 0.036 while DIBR is 0.037. It further demonstrates both methods get pretty good shape recovery. We will add the chamfer comparison in the revision.
>
> >This paper also lacks experimental comparisons with a state-of-the-art method, e.g., [ref1]...
>
> Even though both our application and [ref1] belong to single image inverse graphics method, we want to clarify that [ref1] requires complex lighting supervision while our method, to the best of our knowledge, is the first one that recovers advanced lighting and material without any lighting or material supervision. As an unsupervised method, our approach is quite different from [ref1] and it is unnecessary to compare the two since their supervision is totally different.
>
> >...it would be good to show a system diagram that describes network architecture, network input/output, data flow and training/inference steps
>
> Thanks for the suggestion. We will add a pipeline figure and explain more details of the network in the revision.

---

### Review · Ethics_Reviewer_pEAe · 2021-08-10

**Recommendation:**

The author might want to consider to also report the performance of the method on different subjects in the dataset of human faces, instead of only on the entire dataset or the chosen ones that are not representative for all kinds of subjects.

**Ethical Issues:**

Yes

**Ethics Review:**

The technique proposed in the paper might perform differently for reconstructing human faces, especially considering subjects with different color skins and under various lighting conditions at which the paper targets.

---

> ### Author Response · Authors · 2021-08-18
> **Answers to Reviewer pEAe**
>
> We thank the reviewer for the detailed comments. Our face dataset mostly contains Waucasian faces. We will add more details about face distributions in the dataset and also exame predictions of different subjects of human faces in the revision.

---

### Review · Ethics_Reviewer_9ihu · 2021-08-13

**Recommendation:**

I believe authors can revise and expand the societal implications of their work in the current version---at a minimum by acknowledging that:
* Their method is not tailored to recovering the 3D parameters of human faces and bodies.
* As a result, it should NOT be used in an off-the-shelf manner in settings where such recognition is unjustified (e.g., it leads to privacy harms), or erroneous recognition leads to injury or safety concerns (e.g., a self-driving car failing to properly identify a human on its route).

Additionally, the authors should NOT limit the scope of de-biasing of their tool to including "a wide range of possible skin tones" in the training data. Sources of bias and disparate harms through the use of ML in socially high-stakes domains are vast and they concern not just skin tones, but many other socially salient characteristics of impacted individuals (e.g., the intersection of race and gender). It takes a careful situated analysis to determine if any new method (including the one this paper proposed) is acceptable to use in domains that potentially impact human lives.


**Ethical Issues:**

Yes

**Ethics Review:**

Reviewer 4 raises important concerns about the application of the proposed method to recovering the 3D parameters of human faces.

---

> ### Author Response · Authors · 2021-08-18
> **Answers to Reviewer 9ihu**
>
> We thank the reviewer for the detailed comments. DIBR++ is mostly designed for glossy objects and we will acknowledge that our method is not tailored for humans. We agree that the scope of de-biasing is more than human skin and we will revise the paper to add more discussions about other biases like race and gender.

---

### Author Response · Authors · 2021-08-10
**General Comments**

We thank all the reviewers for their detailed comments. We address shared concerns here and reply to individual comments of each reviewer later.


## Novelty (R1,R2,R4)

How to render with advanced lighting and SVBRDF is not novel in graphics. However, introducing them in learning-based inverse graphics tasks and inferring them from single image in an unsupervised way is not trivial. Previous works can be grouped into three categories:

- Unsupervised 3D reconstruction with naive material and lighting assumption [7,9,10];
- Fully supervised methods relying on either synthetic data or real world data that are expensive to capture [14, ref1, ref2];
- Multiview, optimization-based method [33,34].

Several mentioned previous works, e.g. PhySG [34] and NeRD [33] belong to the third category which only addresses an optimization problem but not learning. [14] and [ref1] can only supervise with synthetic data, making their methods hard to train on real images and may bring domain gap. [ref2] only tackles the subtask of lighting prediction, and trains with data that is costly to acquire. We stress that our method achieves impressive lighting prediction in a fully unsupervised manner without any ground truth supervision, showing strong potential as a valuable source of supervision for the ill-posed task.

In our work, we demonstrate that plausible geometry, lighting, and SVBRDF can be learned from images of realistic scenes (Fig. 6) in an analysis-by-synthesis framework, similar to those found in unsupervised 3D reconstruction. Our work presents an important step towards the design of a practical and general-purpose unsupervised inverse graphics framework. To the best of our knowledge, none of the previous methods demonstrated this before and we are the first to accomplish this goal, which we believe to be a significant contribution.

[ref1] Inverse rendering for complex indoor scenes: Shape, spatially-varying lighting and svbrdf from a single image. CVPR 2020

[ref2] Deep Parametric Indoor Lighting Estimation, ICCV 2019


## Editorial and Visualization Suggestions (R1,R2,R3)
We thank reviewers for their helpful discussions. We will incorporate the editorial suggestions and missing references. We will also add the following visualization figures: the pipeline figure(R1), the GT reference images in different views(R2,R3) and also prepare videos(R2).

---

### Decision · Program_Chairs · 2021-09-28

**Decision:**

Accept (Poster)

**Comment:**

The reviewers found this paper generally well written and interesting, but they had concerns related to novelty, clarity of presentation, and ease of understanding the technical underpinnings of how the model works. The authors addressed many of the concerns and misunderstandings in the rebuttal, but how it looks is that improving upon these points would require a second round of reviews, and thus rejecting the paper in its current form is the right thing to do.

**Consistency Experiment:**

NeurIPS has a long history of experimentation. In 2014, NeurIPS ran an experiment in which 10% of submissions were reviewed by two independent committees to quantify the randomness in the review process. This year, we repeated a variant of this experiment to see how the quality of the review process has changed over time.  This paper was part of the experiment and was therefore assigned to two committees (consisting of reviewers, an Area Chair, and a Senior Area Chair) that reached independent decisions.  If both committees made the same recommendation, this recommendation was followed. If a single committee recommended acceptance, the paper was accepted (with the exception of a few cases in which the other committee identified what we considered a fatal flaw, e.g., an error in a key result).

This copy’s committee reached the following decision: **Reject**

The other committee assigned to the paper recommended **Accept (Poster)**.  You can find the other set of reviews, along with any follow up discussion with the authors here:
https://openreview.net/forum?id=gRqHB07GGz3